# Coevolution of vocal signal characteristics and hearing sensitivity in forest mammals

Benjamin D. Charlton [1], Megan A. Owen [1] & Ronald R. Swaisgood[1]

Although signal characteristics and sensory systems are predicted to co-evolve according to environmental constraints, this hypothesis has not been tested for acoustic signalling across a wide range of species, or any mammal sensory modality. Here we use phylogenetic comparative techniques to show that mammal vocal characteristics and hearing sensitivity have co-evolved to utilise higher frequencies in forest environments – opposite to the general prediction that lower frequencies should be favoured in acoustically cluttered habitats. We also reveal an evolutionary trade-off between high frequency hearing sensitivity and the production of calls with high frequency acoustic energy that suggests forest mammals further optimise vocal communication according to their high frequency hearing sensitivity. Our results provide clear evidence of adaptive signal and sensory system coevolution. They also emphasize how constraints imposed by the signalling environment can jointly shape vocal signal structure and auditory systems, potentially driving acoustic diversity and reproductive isolation.

[1] Institute for Conservation Research, San Diego Zoo Global, California, CA 92027-7000, USA. Correspondence and requests for materials should be addressed to B.D.C. (email: bencharlton829@gmail.com)

A prime objective of animal communication research is to determine the ultimate factors affecting signal characteristics and the sensory capabilities of receivers[1]. Comparative studies of mammal vocal communication have shown that variation in body size[2,3], sexual selection pressures on male calls[2], selection pressures to highlight information encoded by formants[4], and social group size are all drivers of mammal vocal signal diversity[5,6]. Additional comparative work on mammals indicates that the transition to social group living may also drive the evolution of high frequency hearing sensitivity[7]. Another long-standing explanation for the evolution of high frequency hearing in mammals is that it improves the ability to localise sound[8,9] particularly for smaller animals that are more reliant on high frequencies to generate differences in sound intensity reaching the two ears[8]. While the expectation that smaller animals have better high frequency hearing sensitivity is broadly verified across mammal species[10], several exceptions, typically affecting subterranean species, have been documented. These observations indicate that mammal hearing sensitivity is not merely constrained by inter-aural distances but also likely to be driven by other factors, such as the physical environment.

For example, because higher sound frequencies are degraded more rapidly than lower frequencies during atmospheric transmission[11] the ability to perceive low-frequency sound could be important for species that need to communicate over large distances[12,13]. Although lower sound frequencies propagate best in any environment, they are thought to be particularly favoured by selection in acoustically cluttered environments[12–15], such as dense forests, because higher frequencies should be more consistently attenuated than they are in open habitats due to scattering and absorption by stratified media (e.g. branches, leaves, and tree trunks). On the other hand, selection could favour high frequency hearing sensitivity for optimal sound localisation in dense forests with poor visibility, particularly for avoiding predation and localising prey, which would also help to counteract the attenuation of higher frequencies that occurs in forest environments[16–18] and facilitate more effective vocal communication.

The latter contention is plausible because functionally relevant information is often encoded across a wide frequency range in vocalisations, and not just limited to the lower, or lowest frequencies with the most acoustic energy. For instance, a number of mammal studies have shown that formants (vocal tract resonances) have the potential to signal important bio-social information about the caller[19] and these frequency components extend into the upper frequency range (Fig. 1). The dimensions and tissue properties of the supra-laryngeal vocal tract (which comprises the pharyngeal, oral and nasal cavities) determine the formant frequency values and bandwidth in the call spectra[19] (Fig. 1). As a consequence, formants are reliable cues to the caller's body size in a number of species because larger individuals will also have longer vocal tracts that produce lower, more closely spaced formants[20–22]. Other aspects of vocal tract morphology are also likely to differ between individuals, which can result in individually distinctive formant patterns[23–25]. This potentially important information on the identity and size of callers should be present in any call type in which the excitation source adequately highlights the formant pattern[4,24,26,27]. In addition, recent work examining the sound propagation characteristics of mammal vocal signals has revealed that formants are relatively stable and resistant to degradation over distance in forest habitats[28,29] and dense vegetation[30]. These findings indicate that formants could be important for signalling socially relevant information on size and identity in forest environments, where visual cues are often greatly restricted.

Furthermore, according to the Sensory Drive hypothesis[31] mammal vocal characteristics should co-evolve with hearing sensitivity in a predictable direction that is determined by the local signalling environment. Sensory drive posits that the abiotic and biotic environment will influence both signal and sensory systems, which then sets the direction of signal coevolution with respect to receiver sensitivity[31,32]. Support for the complete sensory drive model can be difficult to obtain, however, because it is necessary to reveal a sensory system characteristic that arises through perceptual adaptation to the local environment, and then find signal variation predicted by environmental constraints that corresponds to this sensory bias[31,32]. In addition, to infer underlying evolutionary processes requires comparative tests that take into account the phylogenetic relationships between different species[2,4,7,13,14,33,34]. While this approach has been used to provide good support for coevolution between visual traits and sensory systems[33,35], until now, use of the comparative approach to test sensory drive predictions for vocal communication systems has only considered how vocal characteristics are adapted to transmission in different habitats[13–15,34]. As a result, whether or not the complete sensory drive model applies to vocal communication systems is still an open question.

In this study we use phylogenetic comparative techniques across a wide range of taxa to test the sensory drive hypothesis on mammal vocal communication systems. We find that forest mammals have better high frequency hearing sensitivity when compared to other terrestrial mammals living in more open environments. In line with the sensory drive hypothesis[31], we also show that forest mammals have more high frequency acoustic energy in their vocalisations than other terrestrial mammals, to match hearing sensitivity and optimise the transfer of acoustic information. Finally, for forest mammals with available audiogram and acoustic data we reveal a negative relationship between high frequency hearing sensitivity and high frequency acoustic energy in vocalisations, which suggests forest mammals further optimise vocal communication according to their high frequency hearing sensitivity.

## Results

**Hearing sensitivity versus habitat.** To quantify hearing sensitivity for each species we extracted the frequency of peak hearing sensitivity (in kHz) and calculated the mean hearing threshold values (in dB) for the frequency ranges 0–20 kHz and 10–20 kHz from published audiogram data (Supplementary Table 1). Relative high frequency hearing sensitivity was then calculated by subtracting the mean threshold value for 10–20 kHz from the overall mean (Fig. 2). Phylogenetic generalized linear mixed models (PGLMM) with Bayesian Markov chain Monte Carlo (MCMC) simulations revealed that forest living mammals have higher peak hearing sensitivity than other terrestrial mammals (PGLMM: model effective sample size (ESS) = 1090, phylogenetic heritability $(H^2) = 0.09$, parameter estimate $(\beta) = -3.20$, 95% Credible Interval (CI) = $-5.96$ to $-0.47$, $P_{MCMC} = 0.030$) (Fig. 3a, Supplementary Table 4). $\text{Log}_{10}$ functional head size was not a significant predictor of peak sensitivity (PGLMM: ESS = 1090, $\beta = -1.47$, CI = $-5.55$ to 2.53, $P_{MCMC} = 0.463$) (Supplementary Table 4). Relative high frequency hearing sensitivity was also significantly higher for forest mammals than those living in other terrestrial environments (PGLMM: ESS = 1199, $H^2 = 0.75$, $\beta = -5.82$, CI = $-10.63$ to $-1.17$, $P_{MCMC} = 0.021$) (Fig. 3b) and negatively correlated to $\text{log}_{10}$ functional head size (PGLMM: $\beta = -18.97$, CI = $-27.06$ to $-10.98$, $P_{MCMC} < 0.001$) (Supplementary Table 5). Taken together, these findings indicate that mammal species living in forest environments have better high frequency hearing than those living in more open habitats.

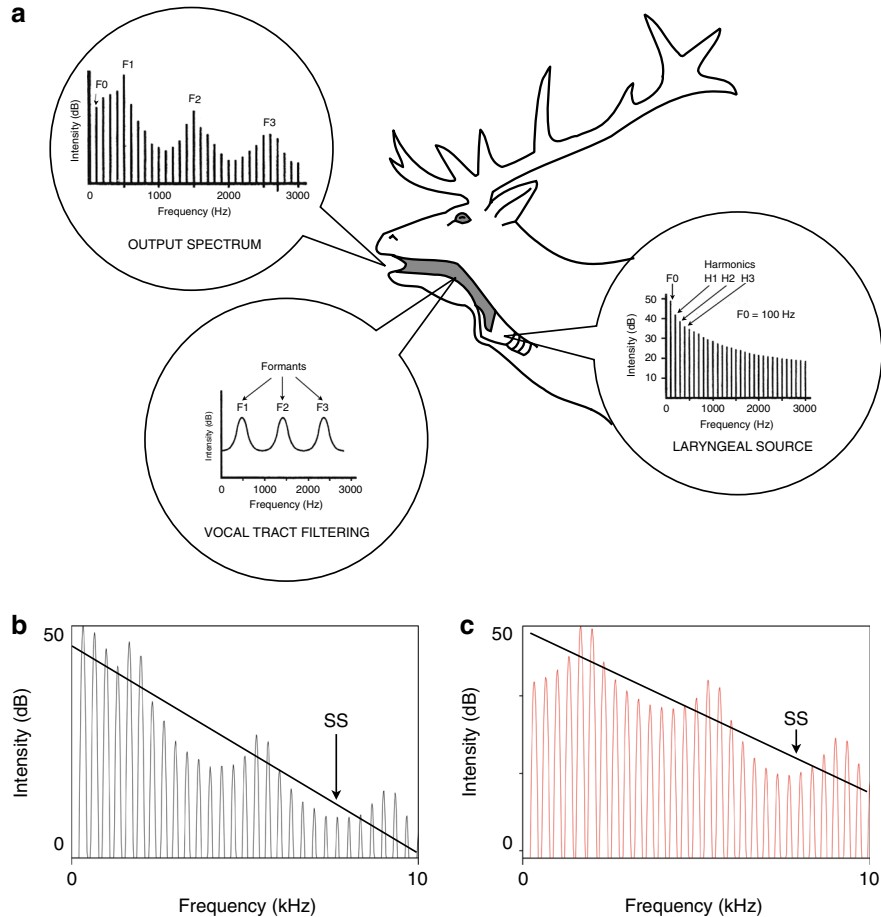

**Fig. 1** Diagrammatic summary of mammal vocal production. Mammal vocalisations typically consist of a source signal that is produced by the larynx and characterised by its fundamental frequency (F0), which corresponds to the rate the vocal folds in the larynx open and close, and a series of harmonic overtones that occur at multiple integers of F0 (labelled H1, H2, H3 etc.) (**a**). The supra-laryngeal vocal tract has its own set of natural resonance frequencies (**a**) that boost the amplitude of certain frequency bands and generate broadband frequency maxima in the sound spectrum termed formants (labelled F1, F2, and F3). The overall shape of the sound spectrum (**a**) is a linear combination of the source signal from the larynx and the filtering effect of the supra-laryngeal vocal tract. The first three formants and underlying harmonic structure of the resultant output spectrum are shown (**a**). Permission to use the red deer stag illustration was kindly provided by Tecumseh Fitch. The lower panels b and c show two sound spectrums with the same F0 (and harmonic spacing) of 100 Hz and the same formant pattern. SS spectral slope (see methods section for details). The red spectrum in panel **c** has relatively more high frequency energy than the spectrum in panel **b**, resulting in a shallower spectral slope. Note that vocalisations with the same F0 and formant patterns can have different spectral energy distributions

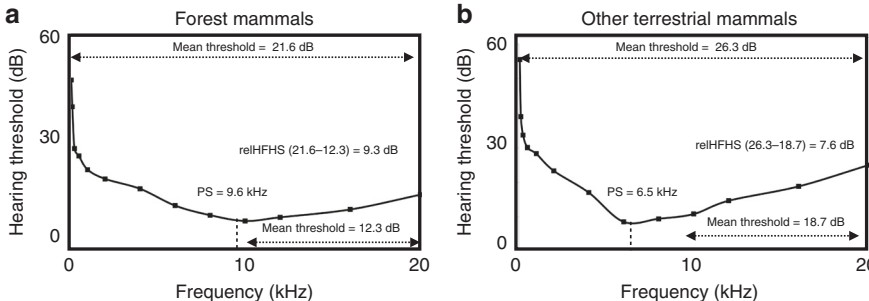

**Fig. 2** Audiogram measures. Composite audiograms derived from the audiogram data for forest mammals (**a**) and mammals that live in other terrestrial habitats (**b**). Audiograms display audible thresholds for tones differing in frequency across a range of hearing, with lower values on the y-axis (dB) indicating greater hearing sensitivity (i.e. lower threshold values). The measures used to characterise the audiograms of different species are shown, and the values presented are derived from the raw data for forest mammals ($n = 24$) and those that typically live in other terrestrial habitats ($n = 27$): relHFHS relative high frequency hearing sensitivity. relHFHS is the mean threshold value for 10–20 kHz minus the overall mean threshold. PS peak sensitivity, the frequency of maximum hearing sensitivity in kHz. Source data are provided as a Source Data file

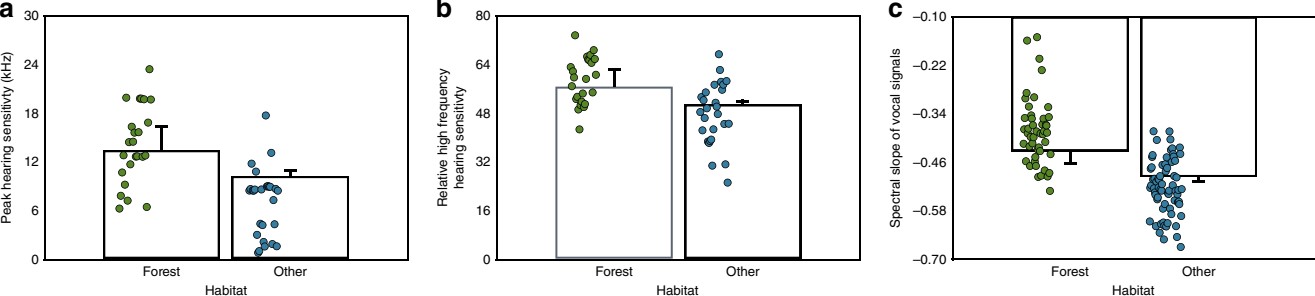

**Fig. 3** Plots of hearing sensitivity and spectral slope versus habitat. Error bar plots of peak hearing sensitivity versus habitat ($N_{species} = 51$) (**a**), relative high frequency hearing sensitivity versus habitat ($N_{species} = 51$) (**b**), and spectral slope versus habitat ($N_{species} = 116$) (**c**). Averaged posterior means + SD taken from three separate MCMC chains are presented. Green and blue circles show the data points for forest and terrestrial mammals, respectively. The PGLMMs that examined peak hearing sensitivity and relative high frequency hearing sensitivity versus habitat included $\log_{10}$ functional head size as a covariate. The PGLMM that examined spectral slope versus habitat included $\log_{10}$ body mass as a covariate and presumed call function as a random factor. The phylogenies used to control for shared ancestry between different species are provided in Supplementary Figs. 1 and 2. Source data are provided as a Source Data file

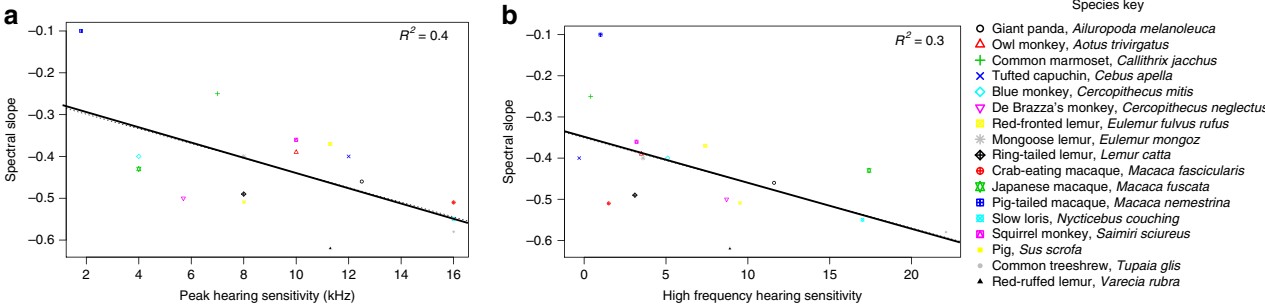

**Fig. 4** Relationship between the spectral slope of forest mammal vocal signals and hearing sensitivity. The raw data are displayed as scatter plots. The solid line represents the average slope and intercept of three MCMC chains from PGLMMs of spectral slope on peak hearing sensitivity (**a**) and high frequency hearing sensitivity (**b**) (both $N_{species} = 17$). $R^2$ values are also provided in the top right-hand corner. The phylogeny used to control for shared ancestry is provided in Supplementary Fig. 3. Source data are provided as a Source Data file

**Spectral energy distribution versus habitat.** To determine whether forest mammals have relatively more high frequency acoustic energy in their vocalisations than other terrestrial mammals, we quantified the frequency distribution of 3701 vocalisations recorded from 116 different terrestrial mammal species by extracting the gradient (slope) of the line connecting spectral peaks, termed the spectral slope[36,37]. Vocal signals with relatively more high frequency acoustic energy will have shallower spectral slopes (Fig. 1). The phylogenetic comparative analysis with MCMC simulations revealed that spectral slopes values were significantly higher for forest mammals than species with more open habitats (PGLMM: ESS = 1090, $H^2 = 0.18$, $\beta = -0.07$, CI = $-0.11$ to $-0.02$, $P_{MCMC} = 0.003$) (Fig. 3c, Supplementary Table 6). $\log_{10}$ body mass was not significantly correlated with spectral slope values (PGLMM: $\beta = 0.00$, CI = $-0.02$ to 0.03, $P_{MCMC} = 0.833$) (Supplementary Table 6). These findings indicate that mammal species living in forest environments produce vocal signals with relatively more high frequency sound energy than those living in other terrestrial habitats.

**Spectral energy distribution versus hearing sensitivity.** Our final phylogenetic comparative analysis with MCMC simulations sought to determine whether forest mammals optimise communication according to their high frequency hearing sensitivity, as predicted by sensory drive[31,32]. For the 17 species with available audiogram and acoustic data we found that peak hearing sensitivity (PGLMM: ESS = 1090, $H^2 = 0.33$, $\beta = -0.02$, CI = $-0.03$

to $-0.00$, $P_{MCMC} = 0.012$) and relative high frequency hearing sensitivity (PGLMM: ESS = 1090, $H^2 = 0.24$, $\beta = -0.01$, CI = $-0.02$ to $-0.00$, $P_{MCMC} = 0.044$) were both negatively correlated with spectral slope values (Fig. 4, Supplementary Tables 7 and 8). These findings indicate that forest species with poorer high frequency hearing sensitivity produce vocal signals with more high frequency acoustic energy.

**Discussion**

In this study comparative analyses were used to test the complete sensory drive model of signal and sensory system coevolution on mammal vocal communication systems. Our phylogenetically-controlled analyses across a wide range of taxa (spanning nine mammalian orders) revealed that forest mammals have greater high frequency hearing sensitivity, and also produce vocal signals with more high frequency acoustic energy than mammals that live in more open habitats. We also revealed that poorer high frequency hearing sensitivity was predictive of shallower spectral slopes in forest mammal vocalisations. These findings suggest that hearing sensitivity and vocal signal characteristics have coevolved: firstly, to facilitate effective communication of information encoded in the upper frequency spectrum in forest environments, and then in the form of an evolutionary trade-off in forest mammals between high frequency hearing sensitivity and the production of calls with high frequency acoustic energy, with decreases in one creating a selection pressure for increases in the other. Although there is some evidence that mammal vocal

signals[13,34] and visual capabilities[38,39] are shaped by the signalling environment, to our knowledge, the results of the current study constitute the first demonstration in mammals of adaptive signal and sensory system coevolution according to environmental constraints.

Our findings also demonstrate that sensory drive[31] applies to vocal communication systems across a wide range of mammals. Comparative work on mammalian auditory anatomy indicates that high frequency hearing is an ancestral trait[40]. Based on this premise, we suggest that it was retained in forest mammals to optimise sound localisation in a visually occluded environment, thereby helping animals to avoid predation, localise prey, and maintain social group cohesion. High-frequency hearing sensitivity could then open up a higher frequency band for auditory communication, leading to vocal signals with more high frequency acoustic energy via sensory drive[31]. Boosting the amplitude of higher frequencies in vocal signals would help to counteract high frequency sound attenuation in forests[17], increase the perceptual salience of formants and any information that they encode, and place less emphasis on lower frequencies for information transfer in an environment with high levels of ambient noise at and below ~3 kHz[12]. Broadband signals with wider frequency spectra are also thought to be easier to locate[41]. Hence, this co-evolutionary process would not only facilitate the transfer of acoustic information encoded by the upper frequency spectrum (i.e. formants), but could also help animals locate vocalising conspecifics in densely forested environments with poor visibility. We suggest that the evolutionary trade-off within forest species reflects a minimal need to provide accurate (i.e. perceivable) information to receivers[42] while avoiding the unnecessary production of more conspicuous signals that may incur a greater risk of predation[41].

Finally, our findings also accord well with recent comparative work that suggests hearing sensitivity in primates is not solely constrained by inter-aural distances[7,43]. We suggest that future studies examine whether mammal vocal signals and sensory systems co-evolve in response to anthropogenic noise, or to utilise frequency ranges that are less likely to be perceived and eavesdropped upon by predators. Sensory drive acting on vocal signals to optimise the transfer of acoustic information in different acoustic environments may prove to be an important driver of mammal vocal signal diversity. It may also explain why forest mammals sometimes produce vocal signals with higher frequency components than closely related species that live in more open habitats[34,44], despite the general consensus that lower frequencies should be favoured in acoustically cluttered habitats[12–15]. Furthermore, given that acoustic signals potentially contribute to reproductive isolation[45], our findings are also consistent with the notion that sensory drive has a wider role in the diversification of mammalian lineages[46]. Future studies should investigate whether acoustic adaptation to forest versus open environments leads to corresponding divergence in mating preferences based on mammal vocal characteristics. A greater understanding of how vocal signal characteristics, auditory perception, and mating preferences based on vocal traits adapt to different local environments will illuminate whether sensory drive contributes to mammal vocal signal diversity and the early stages of reproductive isolation in natural mammal populations.

## Methods

**Audiogram data.** For the comparative analyses of hearing sensitivity we collected audiogram data from the literature for 51 terrestrial mammal species (Supplementary Table 1). Functional head size (defined as the time taken for sound to travel between the two ears) is inversely related to high frequency hearing in mammals[8,9]. It is thought that this inverse relationship exists because low-

frequency sounds (with longer wavelengths) are likely to bypass smaller heads with more closely spaced ears. Smaller species (with smaller heads) are therefore more dependent on higher frequencies for sound localization, and thus, more sensitive to high sound frequencies[8,9]. Accordingly, we took functional head size data from the same source as the audiogram data to control for this factor in the comparative analysis (Supplementary Table 1). We did not collect audiogram data for subterranean species or bats (Supplementary Table 1). Bats were excluded on the basis that they use very high frequency (ultrasonic) echolocation signals to navigate via auto-communication, making it unclear whether their high frequency hearing capabilities are driven by selection pressures linked to navigation or vocal communication, or both. In addition, we restricted the dataset to adult individuals. To maximise our sample size we collected audiogram data generated from behavioural tests ($n = 47$ species) and auditory brainstem responses ($n = 4$ species). Importantly, common parameters of auditory sensitivity, such as the frequency of best sensitivity and the high-frequency limit, are comparable between the two methods[47].

**Audio recordings.** Uncompressed audio recordings (.wav) from 116 terrestrial mammal species were downloaded from the Animal Sound Archive at the Museum für Naturkunde Berlin (http://www.animalsoundarchive.org/) and the Macaulay library at the Cornell Lab of Ornithology (https://www.macaulaylibrary.org), or extracted from commercially available audio CDs (Supplementary Table 2). BDC provided audio recordings for an additional six species (Supplementary Table 2). The original audio recordings had a sampling rate of 44.1 kHz or 48.0 kHz and an amplitude resolution of 16 bits.

We collected recordings from captive animals to avoid examining vocalisations that had already been degraded by environmental transmission in a given species' typical habitat. In addition, only adult vocalisations were collected, and we noted the gender of the animals when it was provided (Supplementary Table 2) so that recordings from both sexes could be obtained whenever possible (53% of species). A minimum of 10 recordings from at least two individuals per taxon was collected. The number of recordings for a given species varied from 10–195 (mean = 32.2) and the number of different individuals contributing to the sample for a given species varied from 2–20 (mean = 3.3). Recordings conducted at different locations were assumed to have originated from different individuals, and those extracted from audio CDs were assumed to originate from only one individual. It must be noted that, although our estimation of individual sample sizes is open to error, if anything it is likely to underestimate as oppose to overestimate the number of potential individuals (since animals at different locations are almost certainly different individuals).

Non-vocal sounds (such as hisses, clicks, rasps, snorts etc.) or low amplitude close range vocal signals (e.g. whimpers, purrs) were not retained for the acoustic analysis, and more than one call type was collected for the majority (64%) of species (Supplementary Table 2). While this helped to remove any potential confound generated by the uneven sampling of calls across species, we also collected data from the literature and recording metadata on the behavioural context of production and presumed function(s) of the different call types in the analysis (Supplementary Table 2). We assigned the different call types to one of the following functional categories: advertisement (mate attraction, territorial), aggression (during or just prior to fighting), alarm (alarm calls), contact (contact promoting calls), disturbance (distress calls, isolation calls), group coordination (recruitment calls, movement calls), and created a 'presumed call function' variable for each species (Supplementary Table 2) to enter as a random factor in the analysis of habitat versus acoustic structure. Species with recordings of vocalisations that spanned more than one of the functional categories were assigned as various.

**Habitat and body weight data.** We determined the typical habitat for each species using information provided by the International Union for Conservation of Nature (IUCN) website (https://www.iucnredlist.org//). All IUCN assessments are peer reviewed by specialists (for more details refer to: https://www.iucnredlist.org/assessment/process). Species stated as occurring primarily in forest environments with no more than three potential habitats were also classed as forest mammals. The other species in our analysis were listed as occurring in grassland, savannah, scrubland, desert, mountain, rocky areas (e.g. inland cliffs and mountain peaks), marine inter-tidal zones, and wetlands (Supplementary Table 3). Because there are strong correlations between body size and the frequency components in mammal vocal signals[2,3] we also collected body mass data (in grams) for each species to control for this factor in the comparative analyses. When body mass data were not available from published studies, we referred to the PANTHERIA v.1 database (Supplementary Table 3).

**Pre-processing of audiogram data and audio recordings.** For each species the frequency of maximum hearing sensitivity (peak sensitivity) in kHz and hearing sensitivity threshold values across a frequency range of 0–20 kHz were extracted from published audiogram data (Supplementary Table 1, Fig. 2). Hearing sensitivity threshold values were restricted to 0–20 kHz because it represented the maximum range that was available for all 51 terrestrial mammal species. Hearing threshold dB values at 20 kHz were estimated by interpolation between adjacent

points (16 kHz and 32 kHz) for 42 species. When peak sensitivity was shared by more than one frequency, the average value was taken (Supplementary Table 1). Additionally, we calculated the mean hearing threshold value for the frequency range 10–20 kHz and subtracted this from the overall mean sensitivity (0–20 kHz), to control for overall differences in hearing sensitivity values across studies (due to methodological differences) and create a standardised index of high frequency hearing sensitivity for each species: higher values indicate better high frequency hearing (Fig. 2).

The audio processing was conducted using Praat v5.1.32 (www.praat.org). Recordings were initially segmented into separate vocalisations using the edit window and labelling facility in Praat and saved as individual sound files (.wav). We discarded recordings with excessive environmental noise, multiple callers with overlapping spectra, and/or sounds other than the targeted vocalisations (e.g., human voices, cage rattling), so that a total of 3701 sound files were retained for the acoustic analysis. All the sound files were down-sampled to 40 kHz, resulting in a Nyquist frequency of 20 kHz that corresponded to our maximum hearing threshold values. The mean intensity of all audio recordings was set to 60 dB prior to the acoustic analysis.

**Acoustic analyses**. To quantify the relative distribution of spectral energy in each of the separate recordings we measured the spectral slope using a Praat (v6.0.31) script from GSU tools[48]. This script computes the spectral slope as a regression line fit to the amplitude peaks of frequency bins across the entire spectrum (for more details refer to supplementary methods). Vocalisations with more high frequency energy will have shallower gradients (or slopes) than those with relatively more low-frequency energy (Fig. 1). The acoustic data was then averaged for each species for the statistical analysis.

**Statistical analysis**. The data were examined using phylogenetic generalized linear mixed models that generated Bayesian posterior probability distributions using Markov Chain Monte Carlo (MCMC) simulations. The Bayesian phylogenetic mixed models were implemented using the MCMCglmm package in R[49,50], with species averaged hearing sensitivity values or spectral slope entered as a Gaussian response variable, habitat (forest or other) as a binary predictor variable, and the phylogenetic relationships among species as a random effect. For the analysis of the audiogram data $log_{10}$ transformed functional head size was also entered as a covariate to control for this factor. For the analysis of the acoustic data we entered $log_{10}$ transformed body mass as a covariate and presumed call function as a random effect to control for these factors. A recent mammal supertree[51] was used to account for common ancestry among species, and pruned prior to each of the three separate analyses to include only species for which we had data (Supplementary Figs. 1, 2 and 3).

For the MCMC simulations we used the default MCMCglmm Gaussian prior with mean = 0 and variance = $10^{10}$ for the fixed effects, and a weakly informative inverse-Gamma prior with shape (alpha) and scale (beta) parameters of 0.001 for random effects. We ran each analysis for 11 million iterations with a burn-in of 100,000 and thinning interval of 10,000 to minimize autocorrelation in the chains. For each model we ran three independent chains (sensu refs. [50,52]) and used the Gelman–Rubin test to ensure model convergence[53]. In all cases a scale reduction factor of one indicated that the chains were indistinguishable and had thus converged (Supplementary Tables 4–8). All the model statistics are reported in Supplementary Tables 4–8, and average values from three separate MCMC chains are reported in the results section. The Heidelberg stationarity test was also used to check for convergence of fixed and random factors within each model (all >0.05) and autocorrelation was checked using trace plots and model outputs (all <0.04 at the first lag). The phylogenetic heritability ($H^2$) was calculated according to Hadfield and Nakagawa[54] using the following equation: $H^2 = \sigma^2_a /(\sigma^2_a + \sigma^2_e)$, where $\sigma^2_a$ is the phylogenetic variance and $\sigma^2_e$ is the residual variance. A $P_{MCMC}$ value of <0.05 was used to denote significant differences between groups or relationships between variables.

**Reporting summary**. Further information on research design is available in the Nature Research Reporting Summary linked to this article.

## Data availability
The data that support the findings of this study are available in the Supplementary Information. The source data underlying Figs. 2, 3 and 4 are provided as a Source Data file.

## Code availability
The R code used for the analyses is available from the corresponding author upon request.

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

## Acknowledgements

We would like to thank Karl-Heinz Frommolt and the Berlin Museum für Naturkunde for providing access to uncompressed recordings.

## Author contributions

B.D.C. wrote the manuscript, collated the data, and conducted the analyses. B.D.C. R.R.S. and M.O. conceived and designed the study. R.R.S. and M.O. contributed to writing the manuscript and collecting the data.

## Additional information

**Competing interests:** The authors declare no competing interests.

