## [Peer Review File · Nature Communications]

Reviewers' Comments:

Reviewer #1:

Remarks to the Author:

I enjoyed this paper very much and find it very interesting. The number of my comments reflects my interest in the paper and should not be taken negatively. All my comments are trivial and aimed at making things clearer.

Line 45, this is an odd sentence because it implies that the physical environment is socio-ecological. That makes no sense. This should be changed to "sociological and ecological factors".

Figure 1. This is interesting, but my immediate reaction is whether or not the slopes of the relationships for forest mammals and other terrestrial mammals are significantly different. The other interesting pattern there is that there is less variation with head size for forest mammals. I suppose that is what needs explaining. You might consider saying more about this very interesting graph.

Figure 2. Since most of the mammals are not large like (red?) deer, it might be good to change this to a mouse or diagrammatic medium to small mammal. Just a thought.

Lines 100-103. But on the other hand I thought that forest environments have far less wind and other air movement noise than open habitats. Please consider discussing this either here or in the discussion. The two countervailing factors make predictions problematic. One other point about localization: In a forest habitat higher frequencies might not be needed for localization because trees make distinct locations so cruder initial targetting, followed by moving to that tree, would be just as efficient as higher frequencies, which might be plagued by high frequency noise from reflections. Again, I hope you will discuss this.

Lines 104-106. That is a very interesting prediction about the possible reciprocal relationship between signal and receiver spectral shapes in sound, I do not think that this prediction has been made previously, although it follows implicitly from sexual selection and communication theory. This makes me think it might be better to make your predictions in a table or as "bullet points" rather than burying them in a single paragraph.

Lines 118-119. What did you do when two models differed in AICc by 2 or less, which is supposed to denote no difference?

Lines 123-124. Why did you calculate the slope of frequency vs hearing thresholds? I thought that a plot of frequency vs hearing thresholds is U shaped or at least has a minimum. In that case a linear relationship is unreasonable. It would have been better to use a quadratic model. Also why did you stop at 20kHz, I thought that some forest mice "sing" well above 20kHz. Or is this just a common upper limit of average equipment and publications? I see from Figure 3 that the slope does seem to capture a lot of the relationship and also that these spectra do indeed go above 20kHz. Did you stop at 20 so that you could use the slope?

Line 131. Break into two paragraphs here; two different subjects.

Lines 137-138. This is a strange way to demonstrate this. It would make far more sense and be a lot more direct to plot the difference between frequency-threshold spectra, and calculate the statistical differences in shape by means of a GAMM (if that can be done phylogenetically; I don't know).

Lines 138-139. Surprising that there was no difference between forest and mixed. Maybe because

mixed has more air movement noise, counteracting the other effects?

Figure 3 needs a caption explaining what you are trying to demonstrate and what a/c vs b/d stand for (forest and open?). One thing I notice about figure 3 is that your slopes do not capture one very different difference between panels a and b: b has a much smaller bandwidth than a, or a narrower range of frequencies. That suggests possibly taking advantage of resonance to transmit and receive efficiently and thereby avoid noise. Anyway, with your measures you are entirely missing bandwidth effects, which might also show interesting patterns.

Figure 4 also needs an explanatory caption; what are the differences between panels a,b and c? The lettering is far too small to read. You might consider making the panels stacked instead of adjacent, to fit the figure into a single journal column.

Line 172. It is fascinating to find two PCs which explained almost equal amounts of variance. However saying 68.1% and 76.6% makes no sense because PCs explanations should add to 100%. I've obviously missed something here! Do these two numbers actually refer to forest and open and are not the first two PCs for a given data set? A better explanation would be a big help.

Line 203, primitive mammals having high frequency sensitivity should not be a surprise because primitive mammals are (and were) small. Worth mentioning?

Lines 207-208. Here you talk about a broader frequency band, but do not test it, even though it appears in figure 2 (if I understand the panels).

Line 209, break paragraph into two here.

Line 227, bad use of "socio-ecological" again. Please revise.

Lines 230-233. This is an excellent suggestion about mixed habitats, and here "bio-social" makes sense but would be better replaced with simply "social". "bio" is redundant.

Lines 238-254. The discussion now seems too long. This paragraph adds very little to the overall point of the paper and can be deleted with no loss. That is the sort of material which belongs in another paper.

Line 257. Another instance of the absurd "socio-ecological". If this is already coagulated in the mammal literature then you had better define exactly what it means the first time you use it. Out of context it makes no sense at all, as I mentioned above.

Line 267-268. The suggestion that sensory drive leads to speciation has been made many times, first by Jenny Boughman and many since her paper (I am not her). You have nicely put a different sensory mode into this context, and this should be made explicit.

Reviewer #2:

Remarks to the Author:

Review of paper: NCOMMS-18-37211

I have reviewed the paper : "Coevolution of vocal signal characteristics and hearing sensitivity in forest mammals" by Charlton et al.

I think this is an excellent, well written paper that investigates a currently neglected area in the field of signal evolution: the coevolution of vocal signal characteristics and hearing sensitivity. Using rigorous, phylogenetically-controlled analyses across 9 orders of mammals, the authors demonstrate that vocal structure and hearing sensitivity have evolved in parallel, likely to make use of higher frequencies in forest environments.

As the authors note, this work has important implications not only for understanding the role of the signaling environment on shaping acoustic communication, but also, more broadly, on the diversification of mammalian lineages, through, for example, acoustic adaptation and subsequent reproductive isolation.

I think the paper will be of significant interest to the readership of Nature Communications and I have just a few comments that the authors might like to consider:

1. My biggest concern regards the way the authors dealt with missing data in the habitat, body weight and mating system data sets. From the methods it seems that, in instances when data were missing, the authors made assumptions regarding these factors, rather than removing these species from the analysis. Whilst I am entirely sympathetic to the drawbacks associated with small sample sizes, I wonder whether it might have been possible to alternatively implement i) some Bayesian statistical approaches which can deal with small sample sizes, or ii) at the least re-running the current models without the species for which data is missing just to check how this impacts the results. I would encourage the authors to consider both alternatives.
2. L176-177: The R squared value here (0.2) seems rather weak. Can the authors comment on this?
3. L428-436: It wasn't totally clear from the statistics section whether the authors controlled for repeated measures at the individual and call level?
4. L311-313: Whilst I appreciate the difficulties associated with extracting caller IDs from pre-recorded data, I think it should still be noted that this approach is open to error.

Reviewer #3:

Remarks to the Author:

This ms tests the hypothesis that signal characteristics and sensory system have evolved in a correlated fashion. To this end, the authors extracted data from the literature and available databases on vocal signal characteristics and hearing sensitivity characteristics for a (variable) sample of mammals that are found in forest, open or mixed habitats. Using standard phylogenetic comparative approaches, the authors show that forest species use and detect higher frequencies. While I find the question somewhat interesting, I have very serious issues on the execution of the study.

METHODS: the authors use acritically a mix of potential evolutionary models without explaining why they picked those models (Brownian motion, PGLS with lambda, rho, OU, OLS). These models make different assumptions about the data and lead to different conclusions, but the choice of model to be tested should be explained and justified. Furthermore, there is no reason to test pure BM, OLS and PGLS with lambda since the latter will return identical results to the former two when lambda is =1 (BM) or =0 (OLS) (see also Freckleton 2009 J Evol Biol). The choice of OU is also not justified and problematic given the small sample size of this study – simulations unambiguously and repeatedly demonstrate that OU models are very prone to identify false positives in support of OU and must not be used with sample sizes of fewer than 200 species (see Cooper et al 2016 Biol J Linn Soc; Ho & Ane' 2014 Methods in Ecology & Evolution; Ho & Ane' 2013 Annals of Statistics). Yet the authors have attempted OU models with multiple predictors and sample sizes as low as 11 species (also see below). The authors mix methods for estimating evolutionary rates (Grafen's rho) with methods for estimating

strengths of evolutionary associations between traits (OU, BM, PGLS with lambda), but the two sets of approaches address different albeit complementary questions (i.e. how quickly traits evolve and how they correlate with other traits). Finally, there are currently an array of much better and more powerful methods than the now outdated Grafen's method to estimate evolutionary rates (e.g. Rabosky et al 2013 Nature Communications; Baker et al 2016 Biol J Linn Soc).

POWER: a major challenge in comparative analyses using data from the literature, is that the sample sizes may vary across variables, so that some analyses may be underpowered. Indeed, in this study sample sizes of key results presented in Table S1 and S3 are small (N=56 and N=11) respectively. Considering that the authors need to estimate intercept and slope for the main predictor, lambda or equivalent parameter depending on the evolutionary model used, this already require at least 30 species (i.e. 10 species per parameter to be estimated). Yet, in some models the authors add 2-4 more predictors (e.g. mating system, sexual size dimorphism, head size, habitat type), with one (habitat) being a discrete 3-level variable requiring on its own 30 species to successfully estimate all parameters in the model. Accordingly, the degrees of freedom in Table S1 vary between 6 and 12, i.e. 60-120 species minimum should be used but only 56 are available. For table S3 df are between 3 and 5, thus a bare minimum of 30 to 50 species are needed but only 11 are available (i.e. not even 2 species are available for computing each parameter!). Therefore, none of the results in Tables S1 and S3, core to the study, can be trusted.

IDENTIFYING THE BEST FITTING MODEL: the most suitable approach, is model reduction (see Crawley's seminal R book) rather than fishing for the 'best' model by running all possible alternative models. The latter approach ends up proposing unrealistic models, i.e. in the supplementary table there are models where predictors are not even close to be significant. Moreover, the authors only present and discuss in the main text the first model, yet at the very least models within 2 AIC score values are equally likely and it is best practice to indeed discuss them all. It is clear from the results tables S1-3 that up to 4-6 alternative models provide equally good fit to the data, but there is no mention of this anywhere in the ms or SI. An additional and major concern is that models in Tables S1-3 include predictors that the authors state are strongly correlated (e.g. SSD and head size); indeed, estimate of size (e.g. head size) are frequently very highly correlated with SSD. Yet the authors do not evaluate to what extent multicollinearity between predictors affects their model (e.g. by first assessing the extent of collinearity with variance inflation factors, and secondly by evaluating its effects on the models), a major concern that can lead to meaningless results if ignored.

DATA QUALITY: the authors use standard principal component analysis to derive their key variables – hearing sensitivity and spectral energy distribution – from multiple correlated measures. However, across species, phylogenetic PCA must always be used instead (Revell 2009 Evolution). A very serious concern is also the quality of the data for the habitat and mating system variables, as these have been extracted from a non-peer reviewed and non-professional website (Animal Diversity Web) maintained by undergraduate students and well known to be often unreliable, containing major errors and inaccuracies. Furthermore, assigning arbitrary values to species with missing data as done by the authors for 5 species with no information on mating system (L343-344) is inventing data! Lastly, the way SSD is computed is incorrect (L333-335). SSD is computed as the log of the ratio of male on female size (which is mathematically equivalent to the difference in Log male minus Log female size), not the ratio of the log male on log female, as incorrectly done here.

WRITING: currently the ms is extremely mammal centric – surely there is lots of information on vocalisation and hearing in other taxa, e.g. birds and insects. Moreover, the ms is currently written for a specialist reader with expertise in vocal communication and hearing, assuming a lot of background knowledge. As such, the ms is better suited for a specialist mammal journal (e.g. Journal of Mammalogy) than for a journal with broad readership.

ADDITIONAL COMMENTS

L29-32: here and throughout, please rephrase this by providing a clear explanation of the underpinning mechanisms, and describing the patterns with direction of effects rather than by simply stating there is an association between traits (in which direction? Why is there one? What does it mean?)

L408: what are 'splitting dates'?

L409: The Bininda-Emonds tree is not a molecular, but instead a supertree.

Fig1 in Introduction: it's unclear where these results come from. This study? How were they computed (phylogenetically)? A fit line on 3 datapoints is meaningless; estimating an intercept, a slope and possibly lambda, with 3 species is simply not meaningful nor useful. The legend provides no detail, including whether the depicted association is significant (likely not for non-forest mammals), nor the implication of this is discussed in the Introduction.

Tables S1-3: The results in the supplementary table should also report the value of the specific model evolutionary parameters, i.e. lambda, alpha for OU, and Grafen's rho. For the OU, it is not clear (indeed not even mentioned) how many adaptive peaks and why the model is set to estimate.

Reviewer #1 (Remarks to the Author):

I enjoyed this paper very much and find it very interesting. The number of my comments reflects my interest in the paper and should not be taken negatively. All my comments are trivial and aimed at making things clearer.

We are very pleased that the reviewer enjoyed the paper and found our results and conclusions interesting.

Line 45, this is an odd sentence because it implies that the physical environment is socio-ecological. That makes no sense. This should be changed to "sociological and ecological factors".

We no longer use the term "socio-ecological".

Figure 1. This is interesting, but my immediate reaction is whether or not the slopes of the relationships for forest mammals and other terrestrial mammals are significantly different. The other interesting pattern there is that there is less variation with head size for forest mammals. I suppose that is what needs explaining. You might consider saying more about this very interesting graph.

This figure was adapted from Heffner & Heffner (In *Handbook of the Senses: Audition*, 2008) to illustrate how functional head size does not appear to explain all the variance in high frequency hearing. We have decided to remove the figure from the manuscript (also in line with one of Reviewer 3's comments) because it is not explicitly linked to the current analysis, and may therefore confuse the reader.

Figure 2. Since most of the mammals are not large like (red?) deer, it might be good to change this to a mouse or diagrammatic medium to small mammal. Just a thought.

We would prefer to leave the figure with a red deer head because it is a study species closely associated with the generalisation of the source-filter theory to nonhuman animals.

Lines 100-103. But on the other hand I thought that forest environments have far less wind and other air movement noise than open habitats. Please consider discussing this either here or in the discussion. The

two countervailing factors make predictions problematic.

Excellent point. Nevertheless, we may expect high frequency sound attenuation to be more consistent in forests where the physical obstacles to sound propagation remain despite prevailing weather conditions. In open habitats high frequencies will certainly be absorbed more readily when it is windy. If it is not windy though, high frequencies are likely to be less attenuated than they would be in a dense forest.

We accept that the statement about forests having stronger/more marked high frequency sound attenuation than open habitats is not always likely to be true. Accordingly, we have now removed “with marked high frequency sound attenuation” from the end of the following prediction (line 105):

“... we also predicted that forest mammals would have more high frequency energy in their vocalisations than other terrestrial mammals, to match hearing sensitivity and optimise the transfer of acoustic information. “

and changed (line 60): “the pronounced attenuation of higher frequencies that occurs in forest environments”

to “the consistent attenuation of higher frequencies that occurs in forest environments”

and changed (lines 55-57): “because higher frequencies are more strongly attenuated due to scattering and absorption by stratified media (e.g. branches, leaves, tree trunks) than they are in open habitats”

to because higher frequencies are likely to be more consistently attenuated than they are in open habitats due to scattering and absorption by stratified media (e.g. branches, leaves, tree trunks)”

One other point about localization: In a forest habitat higher frequencies might not be needed for localization because trees make distinct locations so cruder initial targetting, followed by moving to that tree, would be just as efficient as higher frequencies, which might be plagued by high frequency noise from reflections. Again, I hope you will discuss this.

Good point. High frequency hearing sensitivity may not be that important for localising conspecifics, especially when they reside in certain locations/trees. We do believe, however, that it would be important for avoiding predation and possibly localising prey in a visually occluded forest environment.

We now emphasise this on lines 57-59 of the introduction:

“...selection could favour high frequency hearing sensitivity for optimal sound localisation in dense forests with poor visibility, particularly for avoiding predation and localising prey...”

In the discussion we also postulate that increased high frequency spectral energy in vocal signals may help forest animals locate callers, in line with Marler's predictions (*Nature* 176, 6-8, 1955). To make things clearer here, we now change "locate other conspecifics" to "locate vocalising conspecifics" (on line 212).

Lines 104-106. That is a very interesting prediction about the possible reciprocal relationship between signal and receiver spectral shapes in sound, I do not think that this prediction has been made previously, although it follows implicitly from sexual selection and communication theory. This makes me think it might be better to make your predictions in a table or as "bullet points" rather than burying them in a single paragraph.

We would prefer to keep the predictions in the last paragraph of the introduction.

Lines 118-119. What did you do when two models differed in AICc by 2 or less, which is supposed to denote no difference?

We have now re-analysed the data. In cases where models differed in AIC values by two or less we always select the most parsimonious model with the fewest factors/covariates.

Lines 123-124. Why did you calculate the slope of frequency vs hearing thresholds? I thought that a plot of frequency vs hearing thresholds is U shaped or at least has a minimum. In that case a linear relationship is unreasonable. It would have been better to use a quadratic model.

We no longer use a linear regression slope in our analyses. Instead, we calculate the mean hearing threshold for a frequency range of 10-20 kHz and subtract this from the overall mean (0-20 kHz) to create a standardised index of high frequency hearing for each species (see new Figure 2).

Also why did you stop at 20kHz, I thought that some forest mice "sing" well above 20kHz. Or is this just a common upper limit of average equipment and publications?

Some of the audiograms only go up to 20 kHz. The 20 kHz limit therefore allowed us to compare across all species with available audiogram data.

We explain this on lines 303-305:

"Hearing sensitivity threshold values were restricted to 0-20 kHz because it represented the maximum range that was available for all 51 terrestrial mammal species."

In addition, the mammal vocalizations were recorded using sampling rates of 48 kHz or 44.1 kHz, which will not capture frequencies above 24 or 22.05 kHz, respectively (the Nyquist frequency). To standardize the acoustic

recordings we down-sampled them to 40kHz, giving us the same frequency range for the audiogram and acoustic data (0-20 kHz) in the comparative analyses.

This is explained on lines 319-321:

“All the sound files were down-sampled to 40 kHz, resulting in a Nyquist frequency of 20 kHz that corresponded to our maximum hearing threshold values.”

I see from Figure 3 that the slope does seem to capture a lot of the relationship and also that these spectra do indeed go above 20kHz. Did you stop at 20 so that you could use the slope?

We stopped at 20 kHz to standardize the comparison across all the species with available audiograms (also see response to previous comment).

Line 131. Break into two paragraphs here; two different subjects.

Done.

Lines 137-138. This is a strange way to demonstrate this. It would make far more sense and be a lot more direct to plot the difference between frequency-threshold spectra, and calculate the statistical differences in shape by means of a GAMM (if that can be done phylogenetically; I don't know).

As far as we are aware, it is not possible to conduct a phylogenetically controlled GAMM. We now use an index of high frequency hearing that is derived directly from the hearing threshold values, and standardized within and across species (see Figure 2).

Lines 138-139. Surprising that there was no difference between forest and mixed. Maybe because mixed has more air movement noise, counteracting the other effects?

This effect is no longer apparent with the improved statistical approach and new peer-reviewed habitat data. We also now combine mixed and open into one “other terrestrial mammals” group, in order to increase the sample size for the comparison with forest mammals (see Reviewer 3's comments about sample sizes).

Figure 3 needs a caption explaining what you are trying to demonstrate and what a/c vs b/d stand for (forest and open?).

Part of figure 3 is now presented with the mammal vocal production figure (now Figure 1). We also display composite audiograms (to illustrate the hearing sensitivity measures) as a separate figure (Figure 2). The audiograms are clearly labelled “forest mammals” and “other terrestrial mammals”, and we provide a detailed explanation of what we are trying to demonstrate in the legend.

One thing I notice about figure 3 is that your slopes do not capture one very different difference between panels a and b: b has a much smaller bandwidth than a, or a narrower range of frequencies. That suggests possibly taking advantage of resonance to transmit and receive efficiently and thereby avoid noise. Anyway, with your measures you are entirely missing bandwidth effects, which might also show interesting patterns.

We agree that bandwidth would be very interesting to examine. Unfortunately it was not possible to compare the frequency bandwidths of vocal signals that have a frequency cut off just above 20 kHz (recordings sampled at 44.1 or 48 kHz) with hearing ranges that often extend far above 20 kHz.

In response to this excellent comment we now highlight how boosting the amplitude of higher frequencies would facilitate the transfer of information encoded by formants in the following sentence (lines 204-209).

“Boosting the amplitude of higher frequencies in vocal signals would help to counteract high frequency sound attenuation in forests¹⁷, increase the perceptual salience of formants and any information that they encode, and place less emphasis on lower frequencies for information transfer in an environment with high levels of ambient noise at and below ~3 kHz”

Figure 4 also needs an explanatory caption; what are the differences between panels a,b and c? The lettering is far too small to read. You might consider making the panels stacked instead of adjacent, to fit the figure into a single journal column.

We apologise for the small size of this figure at the review stage. To assist the reader we now include coloured bars that denote forest versus other terrestrial mammals.

Line 172. It is fascinating to find two PCs, which explained almost equal amounts of variance. However saying 68.1% and 76.6% makes no sense because PCs explanations should add to 100%. I've obviously missed something here! Do these two numbers actually refer to forest and open and are not the first two PCs for a given data set? A better explanation would be a big help.

We no longer use a PCA to reduce variables.

Line 203, primitive mammals having high frequency sensitivity should not be a surprise because primitive mammals are (and were) small. Worth mentioning?

Agreed. We have now re-phased this sentence to read (lines 197-199):

“Comparative work on mammalian auditory anatomy and the relatively small size of primitive mammals indicate that high frequency hearing is an ancestral trait”

Lines 207-208. Here you talk about a broader frequency band, but do not test it, even though it appears in figure 2 (if I understand the panels).

We have now changed “broader frequency band” to “higher frequency band” (line 203), which, of course, corresponds to our analyses

Line 209, break paragraph into two here.

Done.

Line 227, bad use of "socio-ecological" again. Please revise.

We no longer use this term.

Lines 230-233. This is an excellent suggestion about mixed habitats, and here "bio-social" makes sense but would be better replaced with simply "social". "bio" is redundant.

The term “bio-social” is no longer used.

Lines 238-254. The discussion now seems too long. This paragraph adds very little to the overall point of the paper and can be deleted with no loss. That is the sort of material which belongs in another paper.

In the interests of brevity and the general readership we have now removed this paragraph, and the discussion of vocal production mechanisms, from the manuscript.

Line 257. Another instance of the absurd "socio-ecological". If this is already coagulated in the mammal literature then you had better define exactly what it means the first time you use it. Out of context it makes no sense at all, as I mentioned above.

Socio-ecological has been removed here and throughout.

Line 267-268. The suggestion that sensory drive leads to speciation has been made many times, first by Jenny Boughman and many since her paper (I am not her). You have nicely put a different sensory mode into this context, and this should be made explicit.

We now cite Boughman (2002) on line 230 and have changed this sentence to read:

“... our findings are also consistent with the notion that sensory drive has a wider role in the diversification of mammalian lineages⁴⁹.”

Reviewer #2 (Remarks to the Author):

Review of paper: NCOMMS-18-37211

I have reviewed the paper: “Coevolution of vocal signal characteristics and hearing sensitivity in forest mammals” by Charlton et al.

I think this is an excellent, well written paper that investigates a currently neglected area in the field of signal evolution: the coevolution of vocal signal characteristics and hearing sensitivity. Using rigorous, phylogenetically-controlled analyses across 9 orders of mammals, the authors demonstrate that vocal structure and hearing sensitivity have evolved in parallel, likely to make use of higher frequencies in forest environments.

As the authors note, this work has important implications not only for understanding the role of the signaling environment on shaping acoustic communication, but also, more broadly, on the diversification of mammalian lineages, through, for example, acoustic adaptation and subsequent reproductive isolation.

I think the paper will be of significant interest to the readership of Nature Communications and I have just a few comments that the authors might like to consider:

We would like to thank the reviewer for these very positive comments.

1. My biggest concern regards the way the authors dealt with missing data in the habitat, body weight and mating system data sets. From the methods it seems that, in instances when data were missing, the authors made assumptions regarding these factors, rather than removing these species from the analysis. Whilst I am entirely sympathetic to the drawbacks associated with small sample sizes, I wonder whether it might have been possible to alternatively implement i) some Bayesian statistical approaches which can deal with small sample sizes, or ii) at the least re-running the current models without the species for which data is missing just to check how this impacts the results. I would encourage the authors to consider both alternatives.

In response to these excellent comments we have: 1) increased the sample size for our third analyses from 11 to 17 species; 2) used Bayesian phylogenetic models in tandem with PGLS models to ensure the robustness of all our results and deal with the relatively small sample sizes; and 3) no longer include species with missing/dummy values.

We also now use peer reviewed habitat data to ensure, as best as we can, that this predictor variable is reliable for the phylogenetic comparative analysis.

2. L176-177: The R squared value here (0.2) seems rather weak. Can the authors comment on this?

The overhaul of the statistical analysis and addition of new data has resulted in much stronger R squared values of 0.4 and 0.3 for the analysis of spectral

slope versus peak hearing sensitivity and high frequency hearing, respectively (see new Figure 4).

3. L428-436: It wasn't totally clear from the statistics section whether the authors controlled for repeated measures at the individual and call level?

Species' averages were used in the analysis. We now make this clear at the end of the acoustic analysis section (lines 330-331):

“The acoustic data was then averaged for each species for the statistical analysis.”

We also state the following in the statistics section (lines 335-337):

“PGLS regressions were conducted using the `pgls` function (`caper` package) in R with species averaged hearing sensitivity values or spectral slope entered as a dependent variable.”

4. L311-313: Whilst I appreciate the difficulties associated with extracting caller IDs from pre-recorded data, I think it should still be noted that this approach is open to error.

We absolutely agree with this comment, and now state the following on lines 276-278:

“It must be noted that our estimation of individual sample sizes is open to error, but more likely to underestimate than overestimate the number (since animals at different locations are almost certainly different individuals).”

Reviewer #3 (Remarks to the Author):

This ms tests the hypothesis that signal characteristics and sensory system have evolved in a correlated fashion. To this end, the authors extracted data from the literature and available databases on vocal signal characteristics and hearing sensitivity characteristics for a (variable) sample of mammals that are found in forest, open or mixed habitats. Using standard phylogenetic comparative approaches, the authors show that forest species use and detect higher frequencies. While I find the question somewhat interesting, I have very serious issues on the execution of the study.

METHODS: the authors use acritically a mix of potential evolutionary models without explaining why they picked those models (Brownian motion, PGLS with λ , ρ , OU, OLS). These models make different assumptions about the data and lead to different conclusions, but the choice of model to be tested should be explained and justified. Furthermore, there is no reason to test pure BM, OLS and PGLS with λ since the latter will return identical results to the former two when λ is =1 (BM) or =0 (OLS) (see also Freckleton 2009 J Evol Biol). The choice of OU is also not justified and problematic given the

small sample size of this study – simulations unambiguously and repeatedly demonstrate that OU models are very prone to identify false positives in support of OU and must not be used with sample sizes of fewer than 200 species (see Cooper et al 2016 Biol J Linn Soc; Ho & Ane' 2014 Methods in Ecology & Evolution; Ho & Ane' 2013 Annals of Statistics). Yet the authors have attempted OU models with multiple predictors and sample sizes as low as 11 species (also see below). The authors mix methods for estimating evolutionary rates (Grafen's rho) with methods for estimating strengths of evolutionary associations between traits (OU, BM, PGLS with lambda), but the two sets of approaches address different albeit complementary questions (i.e. how quickly traits evolve and how they correlate with other traits). Finally, there are currently an array of much better and more powerful methods than the now outdated Grafen's method to estimate evolutionary rates (e.g. Rabosky et al 2013 Nature Communications; Baker et al 2016 Biol J Linn Soc).

We agree entirely with this excellent review point.

In response to this comment we no longer run OU, BM or Grafen models. Instead, we use the caper package in R to conduct PGLS regressions that use maximum-likelihood methods to estimate Pagel's lambda (λ). As the reviewer correctly points out, this then allows us to determine the strength of the phylogenetic signal, with '0' equivalent to a non-phylogenetic (OLS) regression and '1' a pure Brownian motion evolutionary model.

We also justify our use of PGLS and Pagel's lambda (λ) in the text (lines 337-340):

“PGLS models use maximum-likelihood methods to estimate Pagel's lambda (λ), which can then be used to assess the strength of the phylogenetic signal in the model and varies between 0 (phylogenetic independence) and 1 (species' traits co-vary in proportion to their shared ancestry).”

POWER: a major challenge in comparative analyses using data from the literature, is that the sample sizes may vary across variables, so that some analyses may be underpowered. Indeed, in this study sample sizes of key results presented in Table S1 and S3 are small (N=56 and N=11) respectively. Considering that the authors need to estimate intercept and slope for the main predictor, lambda or equivalent parameter depending on the evolutionary model used, this already require at least 30 species (i.e. 10 species per parameter to be estimated). Yet, in some models the authors add 2-4 more predictors (e.g. mating system, sexual size dimorphism, head size, habitat type), with one (habitat) being a discrete 3-level variable requiring on its own 30 species to successfully estimate all parameters in the model. Accordingly, the degrees of freedom in Table S1 vary between 6 and 12, i.e. 60-120 species minimum should be used but only 56 are available. For table S3 df are between 3 and 5, thus a bare minimum of 30 to 50 species are needed but only 11 are available (i.e. not even 2 species are available for computing each parameter!). Therefore, none of the results

in Tables S1 and S3, core to the study, can be trusted.

We have simplified the analysis by reducing the categorical variable habitat to two levels, forest and other mammals, and no longer include mating system, social system or sexual size dimorphism as covariates in a global model (also in line with a subsequent review comment).

In addition, we have increased the sample size of our third analysis from 11 to 17 species, and now run Bayesian phylogenetic models (10,000 iterations) in tandem with the PGLS analyses to ensure the robustness of all of our results (in light of the relatively small sample sizes of $N = 116$, $N = 51$, $N = 17$).

IDENTIFYING THE BEST FITTING MODEL: the most suitable approach, is model reduction (see Crawley's seminal R book) rather than fishing for the 'best' model by running all possible alternative models. The latter approach ends up proposing unrealistic models, i.e. in the supplementary table there are models where predictors are not even close to be significant. Moreover, the authors only present and discuss in the main text the first model, yet at the very least models within 2 AIC score values are equally likely and it is best practice to indeed discuss them all. It is clear from the results tables S1-3 that up to 4-6 alternative models provide equally good fit to the data, but there is no mention of this anywhere in the ms or SI.

Following this excellent point we now choose the simplest model with habitat as a predictor. Head size (for the audiogram data) or body weight (for the acoustic data) are only included when they lower the AIC value by more than 2 units. This has greatly simplified the analysis, and is in line with the model reduction approach suggested by the reviewer.

An additional and major concern is that models in Tables S1-3 include predictors that the authors state are strongly correlated (e.g. SSD and head size); indeed, estimate of size (e.g. head size) are frequently very highly correlated with SSD. Yet the authors do not evaluate to what extent multicollinearity between predictors affects their model (e.g. by first assessing the extent of collinearity with variance inflation factors, and secondly by evaluating its effects on the models), a major concern that can lead to meaningless results if ignored.

We agree that SSD is redundant, and no longer include it as a predictor in our analyses. This ensures that no multicollinearity between SSD and either head size or body weight can occur.

The most complicated model now has habitat as a predictor and functional head size as a covariate (which are not correlated with one another). The other models only have habitat as a predictor, since head size or body weight did not lower the AIC value by more than 2 units.

DATA QUALITY: the authors use standard principal component analysis to derive their key variables – hearing sensitivity and spectral energy

distribution – from multiple correlated measures. However, across species, phylogenetic PCA must always be used instead (Revell 2009 Evolution).

We no longer use a PCA approach.

A very serious concern is also the quality of the data for the habitat and mating system variables, as these have been extracted from a non-peer reviewed and non-professional website (Animal Diversity Web) maintained by undergraduate students and well known to be often unreliable, containing major errors and inaccuracies.

We would like to thank the reviewer for this information. We were unaware that the ADW site was potentially so unreliable, and agree without reservation that another source is required for our habitat data.

In response to this comment we now use the International Union for Conservation of Nature (IUCN) website to obtain habitat data (<https://www.iucnredlist.org/>). This source uses data compiled from scientific studies and all contributions are peer reviewed; firstly by a specialist from the relevant IUCN SSC specialist group, and then the Red List Unit staff who conduct quality checks before publication: for more details refer to: <https://www.iucnredlist.org/assessment/process>. It is also important to point out that we now only make a distinction between forest mammals and those that live in more open (i.e. other) terrestrial habitats.

We have updated the data and all supplementary material accordingly. Habitat data was not available for 8 species from IUCN (*Taurotragus derbianus*, *Ovis aries*, *Felis catus*, *Equus caballus*, *Cavia porcellus*, *Canis familiaris*, *Camelus dromedaries*, *Bos taurus*). These species have now been removed from the analyses.

Furthermore, assigning arbitrary values to species with missing data as done by the authors for 5 species with no information on mating system (L343-344) is inventing data!

Mating system is no longer included in the analysis because we fully accept that these data are not always reliable, and often unresolved for different species. Dummy/arbitrary values are therefore no longer used.

Lastly, the way SSD is computed is incorrect (L333-335). SSD is computed as the log of the ratio of male on female size (which is mathematically equivalent to the difference in Log male minus Log female size), not the ratio of the log male on log female, as incorrectly done here.

SSD is redundant and no longer included in the analysis.

WRITING: currently the ms is extremely mammal centric – surely there is lots of information on vocalisation and hearing in other taxa, e.g. birds

and insects.

Because we specialize in mammal communication systems our focus was to reveal sensory drive in this group of animals for the first time. The inclusion of birds, anurans (frogs and toads) and insects would be beyond the scope of a single paper, and we would be departing from our own areas of expertise.

Moreover, the ms is currently written for a specialist reader with expertise in vocal communication and hearing, assuming a lot of background knowledge. As such, the ms is better suited for a specialist mammal journal (e.g. Journal of Mammalogy) than for a journal with broad readership.

We have removed a paragraph on mammal vocal signal production from the discussion, which would have required more specialist knowledge. General biologists should, however, be able to comprehend all of the other concepts. As the first demonstration of sensory drive in mammals, we strongly feel that the results are appropriate for a journal with broad readership, such as Nature Communications.

ADDITIONAL COMMENTS

L29-32: here and throughout, please rephrase this by providing a clear explanation of the underpinning mechanisms, and describing the patterns with direction of effects rather than by simply stating there is an association between traits (in which direction? Why is there one? What does it mean?)

This comment appears to run counter to the previous call for less specialist/technical explanations that are not directly pertinent to the current study. Providing all of this information about the studies cited in the 2nd sentence of the introduction that the reviewer is referring to would, in our opinion, detract from the papers appeal to the broad readership (i.e. it would involve descriptions of vocal production mechanisms and detailed acoustics). We therefore respectfully disagree with this comment, and would prefer to keep the wording as it is.

L408: what are ‘splitting dates’?

These are nodes. We now state this on line 353.

L409: The Bininda-Emonds tree is not a molecular, but instead a supertree.

We thank the reviewer for pointing this out. We now state the following on lines 352-354:

“For the PGLS regressions and BPMs we used untransformed branch lengths and splitting dates (nodes) from a recent mammal supertree⁵⁵.”

Fig1 in Introduction: it's unclear where these results come from. This study? How were they computed (phylogenetically)? A fit line on 3 datapoints is meaningless; estimating an intercept, a slope and possibly lambda, with 3 species is simply not meaningful nor useful. The legend provides no detail, including whether the depicted association is significant (likely not for non-forest mammals), nor the implication of this is discussed in the Introduction.

This figure was adapted from Heffner & Heffner (In *Handbook of the Senses: Audition*, 2008) to illustrate how functional head size does not appear to explain all the variance in high frequency hearing, and was not linked to our analysis. We totally accept that it was confusing for the reader, however, and it has therefore been removed.

Tables S1-3: The results in the supplementary table should also report the value of the specific model evolutionary parameters, i.e. lambda, alpha for OU, and Grafen's rho. For the OU, it is not clear (indeed not even mentioned) how many adaptive peaks and why the model is set to estimate.

We no longer run OU models or use Grafen's rho as an evolutionary parameter. Values for lambda (λ) are reported in the results section, where we also briefly discuss the implications of its value. A couple of examples of our new approach are provided below:

Lines 133-137: "Relative high frequency hearing sensitivity was strongly influenced by phylogeny and also significantly higher for forest mammals than those living in other terrestrial environments (PGLS: $n = 51$, $\lambda = 0.80$, $t = -2.57$, $P = 0.013$; BPM: eff.samp = 1000, post.mean = 66.09, $P = 0.042$)"

Lines 153-157: "The phylogenetic comparative analysis indicates that variation in spectral slope was independent of phylogeny, and revealed that spectral slopes values were significantly higher for forest mammals than species with more open habitats (PGLS: $n = 116$, $\lambda = 0.00$, $t = -4.00$, $P < 0.001$; BPM: eff.samp = 1000, post.mean = 0.008, $P < 0.001$)"

Reviewers' Comments:

Reviewer #2:

Remarks to the Author:

Re-review of MS: NCOMMS-18-37211A-Z

I have re-reviewed the paper by Charlton et al.

I think the authors have done a great job at dealing with my original comments and concerns in addition to those points raised by the other two reviewers. The MS is particularly strengthened by the inclusion of additional data, the implementation of Bayesian analyses and the switch to a peer-reviewed database for habitat data. I think the paper is now acceptable for publication in Nature Communications.

Minor comments

L151: Should it not be "with relatively higher frequency acoustic energy.."?

L199: Full stop is missing.

L276-278: For readability, perhaps change this to: "It must be noted that, although our estimation of individual sample sizes is open to error, if anything it is likely to underestimate as oppose to overestimate the number of potential individuals (since animals at different locations... etc)".

Reviewer #3:

Remarks to the Author:

The authors have gone to a great length to address many of the issues I raised in my earlier review. Specifically, they have deleted analyses on unjustified or problematic evolutionary models for their sample sizes, slightly increased some sample sizes (specifically for the most problematic analysis now with $n=17$ instead of 11), substantially simplified the models with relevant variables, recollected the habitat data from professional sources and improved the clarity of presentation. As a result, the revised ms has substantially improved. However, there are still areas with serious concerns that need to be addressed.

DATA: Supplementary table 2 shows that there are remarkable differences between the species studied here in the vocalisations analysed and biases across species in the nature of call types, not just sample sizes of individuals. For example, for the red deer we have male roars (presumably during the rut) while for fallow deer female and male moans and groans, and for sika deer only male moans; for other species (e.g. Eulemur, Rucervus) only alarm calls or several vocalisation types. Since individuals use different vocalisations in different behavioural contexts, and different vocalisations should have different acoustic structure, I find problematic that the nature of the calls vary so dramatically across species in this study, and concerning that there is no mention of this bias and how it affects the results anywhere in this ms. While collecting the same call types for many species is difficult, (i) these issues should be discussed in the ms and (ii) some serious effort should be made to evaluate how much this problem may affect the results and conclusion of the study.

We have also no details on how much within species variation compares to across species variation given differences in sample size of individual vocalisations and call type. Given that the authors are now also including analyses in a Bayesian framework, they could easily incorporate within species variation (as additional random effect, easily done in MCMCglmm).

STATISTICAL ANALYSIS: It is not clear in the ms why the authors have chosen to run PGLS in caper (which uses maximum likelihood) and GLMM models in a Bayesian framework in MCMCglmm – from the reading of the response it seems that this might be to evaluate robustness of the results in the light of small sample size of (some) analyses. However, Bayesian statistics faces the exact same issues and difficulties with small sample sizes as any other method; it is not the magic bullet for the lack of data. I have no problem in doing both PGLS in ML and GLMM in MCMC, but the justification provided in the response seems to suggest MCMC as a solution for the lack of data, which is not.

There are several inaccuracies on the description and presentation of the Bayesian models. The definition in the ms of 'Bayesian phylogenetic models' is vague - What is this 'BPM' of?? Is this a GLMM, a probit model, a poisson model or what else? From the incomplete description in the methods it seems that the authors fitted phylogenetic linear mixed models (GLMM) in a Bayesian framework using the package MCMCglmm. These models are not any different conceptually to what PGLS does in that they are both linear (mixed) models with phylogeny (the random effect); they differ in that PGLS in caper estimates the parameters using maximum likelihood while MCMCglmm does so in a Bayesian framework (Markov Chain Monte Carlo). I provide further details below on specific areas of inaccuracies. More concerning, though, is the very short chain and sampling period for the Bayesian analysis as this typically leads to very poor mixing and performance (see detailed comments below).

Note also that from models in MCMCglmm you *must* also report the output of heritability, h^2 , which is how the phylogenetic signal is these models (see Hadfield & Nakagawa 2010 J Evol Biol 23, 494-508), including its ESS (and the ESS, mean and 95% credible intervals for all parameters). Furthermore, if the aim of the authors is to show that PGLS and MCMCglmm produce similar results, then the output must be presented in comparable way by providing the Beta estimates: i.e. for PGLS beta value (and SE) of each model parameter and from MCMCglmm the corresponding posterior beta mean value (and 95% credible interval or SD of the posterior distribution) of the same parameters, alongside p-value/pMCMC, lambda etc.

Finally, MCMC analyses should also be repeated 3-5 times to ensure that the results converge on similar solutions, and more details in needed to explain how convergence and proper mixing were evaluated.

WRITING: I find the presentation of the hypotheses in the Introduction is not fully reflected in the final paragraph where only the predictions of some hypotheses are stated but not those of another hypothesis making opposite predictions; moreover, some predictions cannot be traced back clearly to any hypothesis (see below).

SPECIFIC COMMENTS

L15-17: you present an alternative hypothesis on how cluttered environments may affect the evolution of vocal communication (L57-61), yet it is missing here. Since this is the essence of the study, I would be far clearer in the abstract about testing 2 hypotheses that makes opposite predictions and explain them very briefly here.

L17-19: rather than simply describing the patterns (a tradeoff) please provide also some biological interpretation of what this implies/suggests.

L43: please add here a few words to explain what interaural differences are, also considering that they are repeatedly mentioned in this first paragraph. Figure 1 is very helpful for a non-expert reader and should be presented very early on in the ms.

L67: is identity of the caller detectable in all type of calls or just a specific types of call (i.e. in the context of male-male competition?)

L94: Is this acoustic adaptation hypothesis the same as presented at L50-57? Please clarify here or earlier as appropriate

L103: but you also presented an hypothesis that makes the opposite prediction (L50-57); why is it not even mentioned here?

L107-8: This prediction comes out of the blue, not being mentioned anywhere in the Introduction. As a result it is not clear how and why such prediction is made here and what it implies from a biological point of view. This should be clarified and explained earlier in the Introduction.

L120-121: this makes no sense – PGLS is a statistical approach to account for the phylogenetically structured errors in statistical models, while Bayesian framework (generally MCMC) is a method to estimate model parameters. So PGLS can be done in maximum likelihood or in a Bayesian framework, and a Bayesian model can be a linear model (as presumably here) or lots of other models.

L129-130: what does this mean? Did you estimate the magnitude of the phylogenetic signal in peak hearing sensitivity on its own? If so present the lambda values. Please also note that, while the phylogenetic signal of a single trait may be low (and not statistically different from 0, there may still be a high phylogenetic signal in models including other variables – see Freckleton et al 2002 Am Nat; Revell 2010 Methods in Ecology and Evolution).

L131: Please state clearly that habitat was treated as binary – forest vs non-forest species

L132: what you are showing here is a lambda=0 for the *residuals* of your model, i.e. peak sensitivity across environment, *not* peak sensitivity as such. So, again at L134 it is the residual of sensitivity relative to habitat that are phylogenetically structured, not sensitivity per se. The same interpretation error is perpetuated across the ms, e.g. L154, L168-9

L132-3: What is this BPM of?? What variables? The effective sample size (ESS) needs to be over 1000 for every model parameter estimate (i.e. all fixed effect including the intercept and the random effect, i.e. the phylogeny/heritability), so which one is this referring to here? Equally, what are this mean and p-value of? For MCMC analyses here and throughout you should report 95% credible interval and means for all estimated model parameters. Finally, there is no p-value in Bayesian framework; if anything a pMCMC (although may purists in Bayesian statistics will cringe still at the mention of any p-value).

L137-139: is this the same model as presented at L133-7? If so be clearer on this, if not, how does the statistical effect of habitat change when head size is also included in the model?

Figure 4: I'm afraid I strongly dislike this figure where real data cannot be properly seen as they are covered by the silhouettes of the species. Please show the actual data

L245-6: why head size is a confounding factor must be clearly explained

L344-5: robust to what?

L348: I'm afraid this makes no sense – the posterior is the output of MCMC analysis. I assume that if

you are trying to repeat the analysis in MCMC, what you are doing is a linear mixed model (as in PGLS) but this is very unclear as described here (see my several comments earlier). If so, I assume that what you are trying to say here is that you had used a weakly informative prior, i.e. one derived from a normal distribution with mean of 0 and a large variance (you *must* state the mean value used and the value of 'the large variance'), for the fixed effects. You *must* also report the prior for the random effect, i.e. the phylogeny.

L350-351: I am seriously concerned by this very short chain and extremely short sampling period because the chain can still show poor mixing (there is no information here to evaluate if the authors checked for this nor whether the analyses were repeated and multiple chains converged on similar solutions or not). Typically, the sampling period is at least 1000, often longer (10,000, or hundreds of thousands); a sampling period of only 10 suggests that there are likely issues of mixing in the chain and results may not be reliable. Likewise, the burnin is generally over 10,000, generally 100,000.

Dear Dr. Jones,

The manuscript has now been revised in line with all of the review comments. We would like to thank the reviewers for their constructive comments, and genuinely believe that the paper is now very much improved.

Our responses follow each of the Reviewers' comments (in bold):

Reviewers' comments:

Reviewer #2 (Remarks to the Author):

Re-review of MS: NCOMMS-18-37211A-Z

I have re-reviewed the paper by Charlton et al. I think the authors have done a great job at dealing with my original comments and concerns in addition to those points raised by the other two reviewers. The MS is particularly strengthened by the inclusion of additional data, the implementation of Bayesian analyses and the switch to a peer-reviewed database for habitat data. I think the paper is now acceptable for publication in Nature Communications.

Minor comments

L151: Should it not be “with relatively higher frequency acoustic energy..”?

Yes it should. This has now been changed to (lines 178-9):

“Vocal signals with relatively higher frequency acoustic energy will have shallower spectral slopes...”

L199: Full stop is missing.

A full stop has now been added.

L276-278: For readability, perhaps change this to: “It must be noted that, although our estimation of individual sample sizes is open to error, if anything it is likely to underestimate as oppose to overestimate the number of potential individuals (since animals at different locations... etc)”.

We have now change this sentence to read (lines 322-3):

“It must be noted that, although our estimation of individual sample sizes is open to error, if anything it is likely to underestimate as oppose to overestimate the number of potential individuals...”

Reviewer #3 (Remarks to the Author):

The authors have gone to a great length to address many of the issues I raised in my earlier review. Specifically, they have deleted analyses on unjustified or problematic evolutionary models for their sample sizes, slightly increased some sample sizes (specifically for the most problematic analysis now with $n=17$ instead of 11), substantially simplified the models with relevant variables, recollected the habitat data from professional sources and improved the clarity of presentation. As a result, the revised ms has substantially improved. However, there are still areas with serious concerns that need to be addressed.

DATA: Supplementary table 2 shows that there are remarkable differences between the species studied here in the vocalisations analysed and biases across species in the nature of call types, not just sample sizes of individuals. For example, for the red deer we have male roars (presumably during the rut) while for fallow deer female and male moans and groans, and for sika deer only male moans; for other species (e.g. Eulemur, Rucervus) only alarm calls or several vocalisation types.

Note that across species differently named calls will not necessarily have different acoustic structure. For example, fallow deer 'groans', saiga antelope "roars", and bison 'bellows' are all pulsatile calls with very similar acoustic structure. Call types with the same name will also not necessarily have the same acoustic structure, even within a given mammalian order e.g. red deer roars and musk ox roars (both Artiodactyls).

Since individuals use different vocalisations in different behavioural contexts, and different vocalisations should have different acoustic structure, I find problematic that the nature of the calls vary so dramatically across species in this study, and concerning that there is no mention of this bias and how it affects the results anywhere in this ms. While collecting the same call types for many species is difficult, (i) these issues should be discussed in the ms and (ii) some serious effort should be made to evaluate how much this problem may affect the results and conclusion of the study.

In response to these concerns we have now collected data on the behavioural context of production and purported function(s) of the different call types in the analysis (see updated Supplementary Table 2). The behavioural contexts were taken from the literature for 97 species and the recording metadata for 19 species. This then allowed us to assign the different call types to one of the following functional categories: advertisement (mate attraction, territorial), aggression (during or just prior to fighting), alarm (alarm calls), contact (contact promoting calls), disturbance (distress calls, isolation calls), and group coordination (recruitment calls, movement calls), and create a 'presumed call function' variable for each species to include in the analysis of habitat versus the spectral energy distribution in calls (spectral slope). Species with recordings spanning more than one context or individual call types produced in multiple contexts (e.g. gibbon songs) were labelled as 'various'.

Presumed call function was not a significant predictor of spectral slope: all $P_{MCMC} > 0.1$, and 95% credible intervals (CI) all cross zero. Here are the results:

Aggression,	$\beta = -2.16 \times 10^2$, CI = -9.21×10^2 to 4.92×10^2 ,	$P_{MCMC} = 0.550$
Alarm,	$\beta = -4.89 \times 10^2$, CI = -1.27×10^1 to 3.25×10^2 ,	$P_{MCMC} = 0.232$
Contact,	$\beta = 2.61 \times 10^3$, CI = -7.33×10^2 to 7.88×10^2 ,	$P_{MCMC} = 0.954$
Disturbance,	$-\beta = 7.70 \times 10^2$, CI = -1.84×10^1 to 2.06×10^2 ,	$P_{MCMC} = 0.138$
Group coordination,	$\beta = -1.19 \times 10^2$, CI = -1.30×10^1 to 9.79×10^2 ,	$P_{MCMC} = 0.826$
Various,	$\beta = -5.98 \times 10^3$, CI = -6.34×10^2 to 5.24×10^2 ,	$P_{MCMC} = 0.832$

This confirms that the overall distribution of acoustic energy in calls does not vary consistently across different functional contexts. We nonetheless retain the 'presumed call function' variable in the analysis of the acoustic data as a random factor to control for its associated variance.

We also now make it clear that the formants, which contribute strongly to the spectral slope (see figure 1), are directly linked to the shape and size of the caller's vocal tract, and this should not vary appreciably across different call types (on lines 77-89):

“For instance, a number of mammal studies have shown that formants (vocal tract resonances) have the potential to signal important bio-social information about the caller¹ and these frequency components extend into the upper frequency range (Fig. 1). The dimensions and tissue properties of the supralaryngeal vocal tract (which comprises the pharyngeal, oral and nasal cavities) determine the formant frequency values and bandwidth in the call spectra¹ (Fig. 1). For example, formants are reliable cues to the caller's body size in a number of species because larger individuals will also have longer vocal tracts that produce lower, more closely spaced formants²⁻⁴. Other aspects of vocal tract morphology are also likely to differ between individuals, which can result in individually distinctive formant patterns⁵⁻⁷. This potentially important information on the identity and size of callers should be present in any call type in which the excitation source adequately highlights the formant pattern^{6,8-10}.”

We have also no details on how much within species variation compares to across species variation given differences in sample size of individual vocalisations and call type. Given that the authors are now also including analyses in a Bayesian framework, they could easily incorporate within species variation (as additional random effect, easily done in MCMCglmm).

We made the (fairly safe) assumption that recordings of vocalisations from different locations originated from different animals. However, we have almost certainly underestimated the number of different individuals that contributed to each of the species averaged acoustic values. It is not possible to retrospectively assign recordings taken from audio CDs and downloaded from sound libraries to individuals with any surety, which is a prerequisite for a valid analysis of within species variation (i.e. variation between individuals).

Please refer to the following part of the text (lines 322-3):

“It must be noted that, although our estimation of individual sample sizes is open to error, if anything it is likely to underestimate as oppose to overestimate the number of potential individuals (since animals at different locations...”

It is also important to point out that our aim was to provide species averaged acoustic values that are broadly representative of each taxa, and not to examine associations between traits within and between species.

STATISTICAL ANALYSIS: It is not clear in the ms why the authors have chosen to run PGLS in caper (which uses maximum likelihood) and GLMM models in a Bayesian framework in MCMCglmm – from the reading of the response it seems that this might be to evaluate robustness of the results in the light of small sample size of (some) analyses. However, Bayesian statistics faces the exact same issues and difficulties with small sample sizes as any other method; it is not the magic bullet for the lack of data. I have no problem in doing both PGLS in ML and GLMM in MCMC, but the justification provided in the response seems to suggest MCMC as a solution for the lack of data, which is not.

We included the Bayesian models at the request of reviewer 2. Although they do not make up for a lack of data, they should provide more reliable parameter estimates due to the Markov chain simulations (which are now up from 10,000 to 11,000,000).

Because PGLS in caper and MCMCglmm are both phylogenetically controlled linear models that only differ in how the statistics are generated, we now prefer to use the Bayesian approach and omit the (redundant) PGLS analysis.

There are several inaccuracies on the description and presentation of the Bayesian models. The definition in the ms of ‘Bayesian phylogenetic models’ is vague - What is this ‘BPM’ of?? Is this a GLMM, a probit model, a poisson model or what else? From the incomplete description in the methods it seems that the authors fitted phylogenetic linear mixed models (GLMM) in a Bayesian framework using the package MCMCglmm.

We fitted phylogenetic generalized linear mixed models using a Bayesian framework implemented in the R package MCMCglmm. The response variables were modelled using a Gaussian distribution.

A complete description of the Bayesian phylogenetic models is now provided on lines 400-431 of the methods section

These models are not any different conceptually to what PGLS does in that they are both linear (mixed) models with phylogeny (the random effect); they differ in that PGLS in caper estimates the parameters using maximum likelihood while MCMCglmm does so in a Bayesian framework (Markov Chain Monte Carlo). I provide further details below on specific areas of inaccuracies.

We thank the reviewer for confirming that the PGLS and MCMCglmm approaches do not differ conceptually, but only in how the parameter estimates are derived. We now prefer to use the Bayesian approach and omit

the redundant PGLS analyses.

More concerning, though, is the very short chain and sampling period for the Bayesian analysis as this typically leads to very poor mixing and performance (see detailed comments below).

We would sincerely like to thank the reviewer for helping to improve our Bayesian analysis. We now use a considerably longer chain of 11,000,000 iterations, and a much longer sampling period of 10,000 to minimize autocorrelation between samples. The burn-in is 100,000 iterations.

Note also that from models in MCMCglmm you *must* also report the output of heritability, h^2 , which is how the phylogenetic signal is these models (see Hadfield & Nakagawa 2010 J Evol Biol 23, 494-508), including its ESS (and the ESS, mean and 95% credible intervals for all parameters).

We now report the phylogenetic heritability (H^2) in the main text, along with the effective sample size. H^2 was calculated according to Hadfield and Nakagawa (2010) using the following equation: $H^2 = \sigma_a^2 / (\sigma_a^2 + \sigma_e^2)$, where σ_a^2 is the phylogenetic variance and σ_e^2 is the residual variance.

The posterior mean and 95% credible intervals (CI) for the fixed effect parameter estimates are provided in the main text. Mean and 95%CI for all model parameters are provided in supplementary tables 4-8. In addition, figure 2 now shows the averaged posterior means + SD taken from three separate MCMC chains for each analysis of habitat versus hearing sensitivity values or spectral slope.

Furthermore, if the aim of the authors is to show that PGLS and MCMCglmm produce similar results, then the output must be presented in comparable way by providing the Beta estimates: i.e. for PGLS beta value (and SE) of each model parameter and from MCMCglmm the corresponding posterior beta mean value (and 95% credible interval or SD of the posterior distribution) of the same parameters, alongside p-value/pMCMC, lambda etc.

We no longer compare PGLS with MCMCglmm (see previous comments and response). All model output (statistics and estimates) are provided either in the main text or as supplementary material.

Finally, MCMC analyses should also be repeated 3-5 times to ensure that the results converge on similar solutions, and more details in needed to explain how convergence and proper mixing were evaluated.

We now run three MCMC chains for each phylogenetic model and check they converge on similar solutions using the Gelman and Rubin diagnostic (values of '1' indicated that the chains had converged for all models). The Heidelberg stationarity test was also used to check for convergence of fixed and random

factors within each model (all > 0.05) and autocorrelation was checked using trace plots and model outputs (all < 0.04 at the first lag).

We now explain how convergence and proper mixing were evaluated on lines 419-427:

“For each model we ran three independent chains (*sensu*^{50,52}) and used the Gelman–Rubin test to ensure model convergence⁵³. In all cases a scale reduction factor of one indicated that the chains were indistinguishable and had thus converged (Supplementary Tables 4-8). All the model statistics are reported in Supplementary Tables 4-8, and average values from three separate MCMC chains are reported in the results section. The Heidelberg stationarity test was also used to check for convergence of fixed and random factors within each model (all > 0.05) and autocorrelation was checked using trace plots and model outputs (all < 0.04 at the first lag).”

WRITING: I find the presentation of the hypotheses in the Introduction is not fully reflected in the final paragraph where only the predictions of some hypotheses are stated but not those of another hypothesis making opposite predictions; moreover, some predictions cannot be traced back clearly to any hypothesis (see below).

The general consensus that low frequencies should be favoured in forest environments is historical, and not one of our own predictions. To make this clear in the final paragraph we have changed “... despite predictions to the contrary” to “... despite the general consensus that low frequencies should be favoured in forest environments” (on lines 260-1).

We have also made our predictions at the end of the introduction much clearer (see following responses).

SPECIFIC COMMENTS

L15-17: you present an alternative hypothesis on how cluttered environments may affect the evolution of vocal communication (L57-61), yet it is missing here. Since this is the essence of the study, I would be far clearer in the abstract about testing 2 hypotheses that makes opposite predictions and explain them very briefly here.

We are actually testing the sensory drive hypothesis. This hypothesis predicts that a sensory bias generated by the local signalling environment will be matched by signal characteristics. Lower sound frequencies propagate better than higher frequencies in forest and open habitats, so low frequency hearing for vocal communication should be adaptive in both environments when long distance communication is required. In this study we predicted that high frequency hearing should be favoured in forests over more open habitats to optimise sound localisation in an environment where visual cues are greatly restricted.

We now state that lower frequencies propagate best in any environment on lines 56-8:

“Although lower sound frequencies propagate best in any environment, they are thought to be particularly favoured by selection in acoustically cluttered environments¹²⁻¹⁵”

We have also made our predictions at the end of the introduction much clearer (lines 115-30):

“High frequency hearing should be more adaptive for sound localisation in dense forest with poor visibility than open habitats. Consequently, we predicted that forest mammals would have better high frequency hearing sensitivity when compared to other terrestrial mammals living in more open environments. In line with the sensory drive hypothesis¹⁶, we then expected forest mammals to have more high frequency energy in their vocalisations than other terrestrial mammals, to match hearing sensitivity and optimise the transfer of acoustic information. “

L17-19: rather than simply describing the patterns (a tradeoff) please provide also some biological interpretation of what this implies/suggests.

An interpretation of the trade-off is now provided in the abstract (lines 16-20):

“We also reveal an evolutionary trade-off between high frequency hearing sensitivity and the production of calls with high frequency energy within forest species, **which implies fine-scaled signal and sensory system optimisation within this group of mammals.**”

To make room for this addition in an abstract limited to 150 words we have removed “*across nine mammalian orders*” from lines 13-14 and changed “*to show that vocal characteristics and hearing sensitivity in mammals have co-evolved to utilise higher frequencies in forest environments*” to “*to show that mammal vocal characteristics and hearing sensitivity have co-evolved to utilise higher frequencies in forest environments*”.

L43: please add here a few words to explain what interaural differences are, also considering that they are repeatedly mentioned in this first paragraph.

We no longer use the term ‘inter-aural differences’ here. Instead we have changed lines 45-6 from “to generate inter-aural differences in sound intensity” to “to generate differences in sound intensity reaching the two ears”.

The second mention of inter-aural differences should actually have read inter-aural distances, and has now been changed accordingly (line 50).

Figure 1 is very helpful for a non-expert reader and should be presented very early on in the ms.

Figure 1 is now presented earlier.

L67: is identity of the caller detectable in all type of calls or just a specific types of call (i.e. in the context of male-male competition?)

Caller identity should be detectable in all calls with acoustic sources that adequately highlight the formants. The literature shows that formants are reliable cues to identity in sexual calls (e.g. koala bellows¹, fallow deer groans²), alarm calls (e.g. meerkat barks³, red bellied lemur grunts⁴), contact calls (e.g. giant panda bleats⁵), isolation calls (e.g. dog barks⁶), agonistic calls (e.g. rhesus macaque screams⁷), and calls used for group coordination (e.g.

baboon grunts⁸, elephant rumbles⁹).

1. Charlton, B. *et al.* Perception of male caller identity in koalas (*Phascolarctos cinereus*): acoustic analysis and playback experiments. *PLoS ONE* **6**, e20329 (2011). 2. Vannoni, E. & McElligott, A. G. Individual acoustic variation in fallow deer (*Dama dama*) common and harsh groans: a source-filter theory perspective. *Ethology* **113**, 223-234 (2007). 3. Townsend, S., Charlton, B. & Manser, M. Acoustic cues to identity and predator context in meerkat barks. *Anim. Behav.* **94**, 143-149, (2014). 4. Gamba, M., Colombo, C. & Giacoma, C. Acoustic cues to caller identity in lemurs: a case study. *J. Ethol.* **30**, 191-196, (2012). 5. Charlton, B., Zhang, Z. & Snyder, R. Vocal cues to identity and relatedness in giant pandas (*Ailuropoda melanoleuca*). *J. Acoust. Soc. Am.* **126**, 2721-2732, (2009). 6. Yin, S. & McCowan, B. Barking in domestic dogs: context specificity and individual identification. *Anim. Behav.* **68**, 343-355 (2004). 7. Rendall, D., Owren, M. J. & Rodman, P. S. The role of vocal tract filtering in identity cueing in rhesus monkey (*Macaca mulatta*) vocalizations. *J. Acoust. Soc. Am.* **103**, 602-614 (1998). 8. Rendall, D. Acoustic correlates of caller identity and affect intensity in the vowel-like grunt vocalizations of baboons. *J. Acoust. Soc. Am.* **113**, 3390-3402, (2003). 9. McComb, K., Reby, D., Baker, L., Moss, C. & Sayialel, S. Long-distance communication of acoustic cues to social identity in African elephants. *Anim. Behav.* **65**, 317-329, (2003).

We now make it clear that formant-related information should be present in any call type with an adequate acoustic source on lines 80-89:

“The dimensions and tissue properties of the supra-laryngeal vocal tract (which comprises the pharyngeal, oral and nasal cavities) determine the formant frequency values and bandwidth in the call spectra¹ (Fig. 1). For example, formants are reliable cues to the caller’s body size in a number of species because larger individuals will also have longer vocal tracts that produce lower, more closely spaced formants²⁻⁴. Other aspects of vocal tract morphology are also likely to differ between individuals, which can result in individually distinctive formant patterns⁵⁻⁷. This potentially important information on the identity and size of callers should be present in any call type in which the excitation source adequately highlights the formant pattern^{6,8-10}. “

L94: Is this acoustic adaptation hypothesis the same as presented at L50-57? Please clarify here or earlier as appropriate

Essentially yes. Several papers have made predictions about how animals should adapt acoustic signals according to the prevailing environment, and this has subsequently been termed the “Acoustic Adaptation Hypothesis” in the literature.

We have decided to remove the prefix “*Acoustic Adaptation Hypothesis*” from the citation on line 110 because, in our opinion, it is not a discrete hypothesis and should not be labelled as such. We apologise for this oversight.

L103: but you also presented an hypothesis that makes the opposite prediction (L50-57); why is it not even mentioned here?

Sensory drive is predicated on a pre-existing sensitivity that has evolved for non-communication purposes. The most plausible starting point for sensory drive in forest mammals is high frequency hearing sensitivity for sound localisation in a visually restricted environment. This then forms the basis for other predictions that vocal signal structure should co-evolve with hearing sensitivity.

We now make this clear on lines 115-119:

“High frequency hearing should be more adaptive for sound localisation in dense forest with poor visibility than open habitats. Consequently, we predicted that forest mammals would have better high frequency hearing sensitivity when compared to other terrestrial mammals living in more open environments. “

L107-8: This prediction comes out of the blue, not being mentioned anywhere in the Introduction. As a result it is not clear how and why such prediction is made here and what it implies from a biological point

of view. This should be clarified and explained earlier in the Introduction.

This has now been changed to more accurately reflect our original intention to examine the relationship between hearing sensitivity and the spectral energy distribution of vocalisations in forest mammals, to determine whether they further optimise vocal communication according to their high frequency hearing sensitivity. We then interpret the relationship in the discussion.

This section now reads (lines 130-4):

“Finally, for forest mammals with available audiogram and acoustic data, we examined the relationship between hearing sensitivity and the spectral energy distribution of vocalisations. This allowed us to determine whether this group of mammals further optimise vocal communication according to their high frequency hearing sensitivity.”

L120-121: this makes no sense – PGLS is a statistical approach to account for the phylogenetically structured errors in statistical models, while Bayesian framework (generally MCMC) is a method to estimate model parameters. So PGLS can be done in maximum likelihood or in a Bayesian framework, and a Bayesian model can be a linear model (as presumably here) or lots of other models.

We entirely agree with the reviewer, and now only use the Bayesian approach to test our hypotheses.

This section has been reworded accordingly (lines 137-140):

“We used phylogenetic generalized linear mixed models (PGLMM) with Bayesian Markov chain Monte Carlo (MCMC) simulations to test our hypotheses while controlling for the confounding effects of shared phylogenetic ancestry (see methods section for details).”

L129-130: what does this mean? Did you estimate the magnitude of the phylogenetic signal in peak hearing sensitivity on its own? If so present the lambda values. Please also note that, while the phylogenetic signal of a single trait may be low (and not statistically different from 0, there may still be a high phylogenetic signal in models including other variables – see Freckleton et al 2002 Am Nat; Revell 2010 Methods in Ecology and Evolution).

The lambda value was for the entire model i.e. with habitat as a predictor. We no longer run PGLS with lambda (see earlier responses).

L131: Please state clearly that habitat was treated as binary – forest vs non-forest species

This is now clearly stated on line 405 - “habitat (forest or other) as a binary predictor variable,..”

L132: what you are showing here is a lambda=0 for the *residuals* of your model, i.e. peak sensitivity across environment, *not* peak

sensitivity as such. So, again at L134 it is the residual of sensitivity relative to habitat that are phylogenetically structured, not sensitivity per se. The same interpretation error is perpetuated across the ms, e.g. L154, L168-9

We thank the reviewer for pointing this out. Since we only want to control for shared phylogeny in the models (not determine the phylogenetic signal for characteristics) all statements about phylogenetic strength have now been removed. Note also that we no longer use PGLS with lambda.

L132-3: What is this BPM of?? What variables?

We refer the reader to the methods section on lines 139-140, which now contains a complete description of the Bayesian phylogenetic models.

Lines 137-140 now read:

“Phylogenetic Generalised Linear Mixed Models (PGLMM) with Bayesian Markov Chain Monte Carlo (MCMC) simulations were used to control for the confounding effects of shared phylogenetic ancestry while testing our hypotheses (see methods section for more details).”

The effective sample size (ESS) needs to be over 1000 for every model parameter estimate (i.e. all fixed effect including the intercept and the random effect, i.e. the phylogeny/heritability), so which one is this referring to here?

The effective sample size is now over 1000 for all fixed and random effects, and the intercept for each model (see supplementary tables 4-8).

Equally, what are this mean and p-value of? For MCMC analyses here and throughout you should report 95% credible interval and means for all estimated model parameters. Finally, there is no p-value in Bayesian framework; if anything a pMCMC (although may purists in Bayesian statistics will cringe still at the mention of any p-value).

We now report the 95% credible interval and means for all estimated model parameters in supplementary tables 4-8. pMCMC values are used to denote significance (not p values).

L137-139: is this the same model as presented at L133-7? If so be clearer on this, if not, how does the statistical effect of habitat change when head size is also included in the model?

A model selection approach is no longer used. Instead, we now include head size and body mass as covariates in our between subjects analyses of hearing sensitivity and spectral slope, respectively, to control for these factors that are known to influence hearing sensitivity and the acoustic structure of mammal calls. The statistics for both factors are reported in the main text irrespective of statistical significance.

Figure 4: I'm afraid I strongly dislike this figure where real data cannot be properly seen as they are covered by the silhouettes of the species. Please show the actual data

We now show the actual data points with the icons alongside so the reader to relate the data to the different species.

L245-6: why head size is a confounding factor must be clearly explained

We now explain this fully on lines 277-285:

“Functional head size (defined as the time taken for sound to travel between the two ears) is inversely related to high frequency hearing in mammals^{8,9}. It is thought that this inverse relationship exists because low-frequency sounds (with longer wavelengths) are likely to bypass smaller heads with more closely spaced ears, thereby leaving smaller species (with smaller heads) more dependent on higher frequencies for sound localization, and thus, more sensitive to high sound frequencies^{8,9}. We therefore took functional head size data from the same source as the audiogram data to control for this factor in the comparative analysis (Supplementary Table 1).”

L344-5: robust to what?

The term “robust” is now longer used here.

L348: I'm afraid this makes no sense – the posterior is the output of MCMC analysis. I assume that if you are trying to repeat the analysis in MCMC, what you are doing is a linear mixed model (as in PGLS) but this is very unclear as described here (see my several comments earlier).

The statistics section has been completely re-written (lines 400-431). We hope that things are much clearer now.

If so, I assume that what you are trying to say here is that you had used a weakly informative prior, i.e. one derived from a normal distribution with mean of 0 and a large variance (you *must* state the mean value used and the value of ‘the large variance’), for the fixed effects. You *must* also report the prior for the random effect, i.e. the phylogeny.

This has now been done (lines 415-418):

“For the MCMC simulations we used the default MCMCglmm Gaussian prior with mean = 0 and variance = 10^{10} for the fixed effects, and a weakly informative inverse-Gamma prior with shape (alpha) and scale (beta) parameters of 0.001 for random effects. “

L350-351: I am seriously concerned by this very short chain and extremely short sampling period because the chain can still show poor mixing (there is no information here to evaluate if the authors checked for this nor whether the analyses were repeated and multiple chains converged on similar solutions or not). Typically, the sampling period is at least 1000, often longer (10,000, or hundreds of thousands); a sampling period of only 10 suggests that there are likely issues of mixing in the chain and results may not be reliable. Likewise, the burnin is generally over 10,000, generally 100,000.

We now run each analysis for 11 million iterations with a burn-in of 100,000 and thinning interval of 10,000 to minimize autocorrelation in the chains. In addition, for each model we run three independent chains and use the Gelman–Rubin test to ensure model convergence. In all cases a scale reduction factor of one indicated that the chains had converged (Supplementary Tables 4-8). The Heidelberg stationarity test was also used to check for convergence of fixed and random factors within each model (all > 0.05) and autocorrelation was checked using trace plots and model outputs (all < 0.04 at the first lag).

Reviewers' Comments:

Reviewer #3:

Remarks to the Author:

The authors have addressed all my major points satisfactorily. There are however some issues with the presentation of the results, but this hopefully is only due to superficial typos in the text and tables albeit many (see below), than be caused by far more serious problems with the analyses.

L152 and Table S4: there is something wrong with the results as reported. The mean Beta estimate for habitat *cannot* fall outside the 95% CI of its posterior distribution. Similar disconcerting issues are found in many of the values reported in all Supplementary Tables S4-8; these must all be carefully checked and corrected.

Supplementary Tables S4–8: H^2 values are incorrectly reported (or incorrectly computed); H^2 ranges only between 0 and 1, as reported these values are incorrect.

Supplementary Tables S4–8: there are several errors in the reported values (beta mean and CI) across all rows of all tables; please be careful and check when reporting values as $\times 10^x$ vs as $\times 10^{-x}$; the minus matters and seems to be missing in some of the reported values or else nothing here makes any sense!

MINOR COMMENTS

L151 and throughout, and Supplementary tables: effective sample sizes are generally names ESS (not `eff.samp` which is just a MCMCglmm package name). Most importantly, though, you need to be consistent between text (`eff. Samp`) and tables (ES).

L231: 'primitive' mammals being what taxon exactly?

Figure 3 legend and in the text: in \log_{10} , the 10 is subscript not superscript.

L406-7 and 408: 'between subjects' is not appropriate description for a cross-species analysis.

Table 7, controlling for call function type: glad to see the results hold when this is accounted for, but please do mention in the result section of the main text that the models also controlled for call function, as other might also wonder

Dear Dr. Jones,

The manuscript has been revised in line with all of the review comments. Our responses follow each of the Reviewers' comments (in bold):

Reviewer #3 Comments to the authors:

The authors have addressed all my major points satisfactorily. There are however some issues with the presentation of the results, but this hopefully is only due to superficial typos in the text and tables albeit many (see below), than be caused by far more serious problems with the analyses.

L152 and Table S4: there is something wrong with the results as reported. The mean Beta estimate for habitat *cannot* fall outside the 95% CI of its posterior distribution. Similar disconcerting issues are found in many of the values reported in all Supplementary Tables S4-8; these must all be carefully checked and corrected.

We would like to thank the reviewer for noticing these typos that occurred during the conversion to scientific notation. We now use standard numbers expressed to two decimal places.

Supplementary Tables S4—8: H² values are incorrectly reported (or incorrectly computed); H² ranges only between 0 and 1, as reported these values are incorrect.

The H² values were incorrectly converted to scientific notation. We now report H² using standard numbers expressed to two decimal places.

We are aware that the values must all fall between 0-1. H² was correctly calculated using the formula: $H^2 = \sigma_a^2 / (\sigma_a^2 + \sigma_e^2)$, where σ_a^2 is the phylogenetic variance and σ_e^2 is the residual variance.

Supplementary Tables S4—8: there are several errors in the reported values (beta mean and CI) across all rows of all tables; please be careful and check when reporting values as x10^x vs as x10^{-x}; the minus matters and seems to be missing in some of the reported values or else nothing here makes any sense!

These values have all been converted back to standard numbers and reported to two decimal places. All of the table values have also been double-checked.

MINOR COMMENTS

L151 and throughout, and Supplementary tables: effective sample sizes are generally names ESS (not eff.samp which is just a MCMCglimm package

name). Most importantly, though, you need to be consistent between text (eff. Samp) and tables (ES).

We now use ESS to denote effective sample size throughout the manuscript and supplementary material.

L231: 'primitive' mammals being what taxon exactly?

We inserted “and the relatively small size of primitive mammals” in response to an earlier (minor) review comment.

In response to this comment we no longer refer to primitive mammals here, and have reverted the sentence back to its original:

“Comparative work on mammalian auditory anatomy indicates that high frequency hearing is an ancestral trait⁴⁰”

Figure 3 legend and in the text: in Log 10, the 10 is subscript not superscript.

These typos have been corrected.

L406-7 and 408: 'between subjects' is not appropriate description for a cross-species analysis.

We agree - 'between subjects' has now been removed.

Table 7, controlling for call function type: glad to see the results hold when this is accounted for, but please do mention in the result section of the main text that the models also controlled for call function, as other might also wonder

We clearly state that call function was controlled for in the analysis in the Figure 3 legend: “The PGLMM that examined spectral slope versus habitat included \log_{10} body mass as a covariate and presumed call function as a random factor.“

And again in the methods section: “For the analysis of the acoustic data we entered \log_{10} transformed body mass as a covariate and presumed call function (Supplementary Table 4) as a random effect to control for these factors”